# Development of mesothelioma-specific oncolytic immunotherapy enabled by immunopeptidomics of murine and human mesothelioma tumors

Jacopo Chiaro[1,2,3,4,17], Gabriella Antignani[1,2,3,4,17], Sara Feola [1,2,3,4], Michaela Feodoroff [1,2,3,4,5], Beatriz Martins[1,2,3,4], Hanne Cojoc[6], Salvatore Russo[1,2,3,4], Manlio Fusciello [1,2,3,4], Firas Hamdan [1,2,3,4], Valentina Ferrari[7], Daniele Ciampi [7], Ilkka Ilonen [8,9], Jari Räsänen[8,9], Mikko Mäyränpää [10], Jukka Partanen [11], Satu Koskela [12], Jarno Honkanen[12], Jussi Halonen[12], Lukasz Kuryk [6,13], Maria Rescigno [7,14], Mikaela Grönholm[1,2,3,4], Rui M. Branca [15], Janne Lehtiö [15] & Vincenzo Cerullo [1,2,3,4,16] ✉

Malignant pleural mesothelioma (MPM) is an aggressive tumor with a poor prognosis. As the available therapeutic options show a lack of efficacy, novel therapeutic strategies are urgently needed. Given its T-cell infiltration, we hypothesized that MPM is a suitable target for therapeutic cancer vaccination. To date, research on mesothelioma has focused on the identification of molecular signatures to better classify and characterize the disease, and little is known about therapeutic targets that engage cytotoxic (CD8+) T cells. In this study we investigate the immunopeptidomic antigen-presented landscape of MPM in both murine (AB12 cell line) and human cell lines (H28, MSTO-211H, H2452, and JL1), as well as in patients' primary tumors. Applying state-of-the-art immuno-affinity purification methodologies, we identify MHC I-restricted peptides presented on the surface of malignant cells. We characterize in vitro the immunogenicity profile of the eluted peptides using T cells from human healthy donors and cancer patients. Furthermore, we use the most promising peptides to formulate an oncolytic virus-based precision immunotherapy (PeptiCRAd) and test its efficacy in a mouse model of mesothelioma in female mice. Overall, we demonstrate that the use of immunopeptidomic analysis in combination with oncolytic immunotherapy represents a feasible and effective strategy to tackle untreatable tumors.

Malignant pleural mesothelioma (MPM) is a highly aggressive form of cancer with a poor prognosis, characterized by a median survival rate of less than a year[1,2]. This cancer primarily occurs due to exposure to asbestos fibers and is commonly classified as an occupational disease[3].

Despite the ban on asbestos in numerous countries, it continues to be extensively used in many developing nations. One major challenge is that tumors could develop even many years after asbestos exposure. The current treatment options for MPM include chemotherapy,

radiotherapy, and surgery, which have shown limited effectiveness to date. The incidence of MPM is progressively rising, highlighting the urgent requirement for novel therapies and treatment targets[1].

Tumors with a high infiltration of T cells, known as "hot" tumors, are generally associated with a favorable prognosis. Although MPM is not universally categorized as a "hot" tumor, it does exhibit infiltration by T cells, making it a potential candidate for the use of immunotherapies[4]. As immune checkpoint inhibitors (ICIs) have demonstrated remarkable efficacy in other solid malignancies, their use as MPM treatment has increasingly been explored in clinical trials. However, ICIs as monotherapies have had limited impact on the overall survival of MPM patients[3].

Although other kinds of immunotherapies offer promise in ongoing clinical trials for MPM, they are still in their early stages. Consequently, there is an urgent need for further research to propose innovative treatment strategies for MPM[3,5]. One such innovative therapeutic option could involve the use of Oncolytic viruses (OVs). These viruses are genetically modified to selectively replicate in and eliminate cancer cells. Additionally, OVs stimulate the immune system, inducing a more immunogenic tumor microenvironment and potentially increasing the effectiveness of other cancer treatments[3].

To enhance the immune response against the tumor, our laboratory has developed a technology called PeptiCRAd[6]. PeptiCRAd involves the use of an oncolytic adenovirus (OAd) coated with tumor-specific peptides, combining viral immunogenicity with the cancer specificity of the peptides. Previous studies have demonstrated the efficacy of PeptiCRAd in vivo for various tumor types[7–11].

The identification of molecular signatures to classify the disease into subtypes and stratify patients based on genomic analysis has been the focus of prior research efforts in MPM[12,13]. Consequently, the number of tumor antigens considered for MPM remains limited. Few tumor antigens are under investigation, such as Mesothelin (*MSLN*) for CAR-T cell therapy and Wilms Tumor antigen (*WT1*) for therapeutic cancer vaccines. However, the effectiveness of these antigens in treating MPM is still being evaluated[3,14]. Therefore, it is crucial to identify novel targetable antigens for this cancer type.

In contrast to transcriptomics and proteomics techniques that rely on in-silico predictions of antigens presented on the tumor cell surface, direct immunopurification of the MHC-antigen complex is the most effective approach for identifying potential CD8+ T cell targets[15]. Thus, we study here the synergistic effect of immunopeptidomics-identified peptides in combination with viral immunotherapy using the PeptiCRAd vaccinal platform.

In this study, we investigate the MHC-I antigen landscape of MPM. We analyze the immunopeptidome of both murine (AB12 cell line) and human MPM cell lines (H28, MSTO-211H, H2452, and JL1), as well as patients' tumor samples, using state-of-the-art MHC complex immunoprecipitation and mass spectrometric methodologies. Moreover, as MPM is a tumor type characterized by a low mutation burden[16], we focus on the use of tumor-associated antigens (TAAs) as search space for identifying novel targets for the development of cancer vaccines in mesothelioma. Thorough analysis of the eluted peptide datasets, using our in-house script (PyptidOmicsQC), demonstrates qualitative alignment with previously published immunopeptidomics datasets of other tumor types, validating our methodology and confirming the identification of relevant MHC-I antigens.

Importantly, peptides identified through immunopeptidomics of human cell lines and patient-derived tumor samples display promising immunogenicity profiles. These peptides are capable of activating CD8+ T cells and promoting the killing of mesothelioma cell lines, highlighting their potential as targets for the development of immunotherapies to treat MPM.

In the second part of our study, we provide a proof-of-concept for the entire vaccine-development pipeline, starting from antigen discovery to the production of a therapeutic cancer vaccine, evaluating its efficacy in a murine model of MPM. We begin by investigating the immunopeptidome of the AB12 murine mesothelioma cell line. After performing in silico and functional characterization of the eluted peptides, we use the most promising candidate peptides to generate a proof-of-concept oncolytic cancer vaccine for MPM using the PeptiCRAd platform.

Our in vivo data demonstrate that immunogenic peptides identified via immunopeptidomics are tumor-specific. Intratumoral administration of PeptiCRAd coated with the AB12-derived immunogenic peptides produces a more immunogenic tumor microenvironment and induce a potent anti-tumor immune response, resulting in an improved control of AB12 tumor growth in tumor-bearing mice.

This study expands the repertoire of therapeutic targets for MPM and demonstrates the potential of immunopeptidomics for developing effective cancer vaccines.

## Results
### The immunopeptidomic profile of mesothelioma is a representation of MHC-I alleles' composition
Mesothelioma has been extensively studied to establish its molecular fingerprint for systematic classification. However, a comprehensive understanding of the antigen landscape required for the development of targeted immunotherapy is still lacking. To address this, we conducted an immunopeptidomic analysis of both murine (AB12 cell line) and human MPM cell lines (H28, MSTO-211H, H2452, and JL1), as well as patients' tumor samples.

We first assessed the expression levels of MHC-I molecules on the surface of the human mesothelioma cell lines using flow cytometry, which confirmed high MHC-I expression in all the cell lines (Supplementary Fig. 1A). These findings confirmed the suitability of all the mesothelioma cell lines for immunopeptidomic analysis.

The MHC-I binding peptides were eluted using state-of-the-art immunoaffinity purification and subsequently characterized by LC-MS/MS using a human canonical proteome as a reference database for spectral matching (Fig. 1a).

For the human cell lines, we performed multiple biological replicates: H28 (*n* = 3), MSTO-211H (*n* = 6), H2452 (*n* = 4), and JL1 (*n* = 2). As a result of limited sample availability, only one replicate was performed for patient MESO001, whereas three replicates were conducted for patient MESO002, comprising two from tumor tissues and one from adjacent "benign" tissue. The immunopeptidomic data obtained from the run of the MESO002 samples revealed an abnormal quantity of long eluted peptides, which were identified as contaminants. To address this issue, the dataset underwent in silico curation prior to any analysis (For more information, please refer to the "Methods" section: MESO002 immunopeptidome data cleanup). Across all the cell lines, we observed a higher number of eluted peptides for the MSTO211H and H2452 cell lines. 8-13mers accounted for ~83% of all eluted peptides (highest for the 211H with 91%). Among these, 9mers represented around 50% of the 8-13mers interval and ~40% of the overall eluted peptides. Human mesothelioma tumor resections yielded a different number of unique peptides, with over half of them falling within the 8-13mers interval (64%). Around 40% of the 8-13mers interval and ~25% of the total peptides were 9mers (Supplementary Fig. 1B).

To ensure the quality of our immunopeptidomics datasets, we developed an in-house script called PyptidOmicsQC, which automates a series of quality control assessments for immunopeptidomics experiments. Using this tool, we verified that the length distribution of the eluted peptides exhibited the expected peak at nine amino acids in length for all analyzed samples (Fig. 1b). Furthermore, using state-of-the-art machine learning-based methods for peptide MHC binding affinity prediction, we observed that the majority of the peptides eluted in our replicates were predicted to be MHC binders. On average, 83% of the eluted peptides within the 8-13 amino acids range were found to be specific for the corresponding MHC haplotype. When

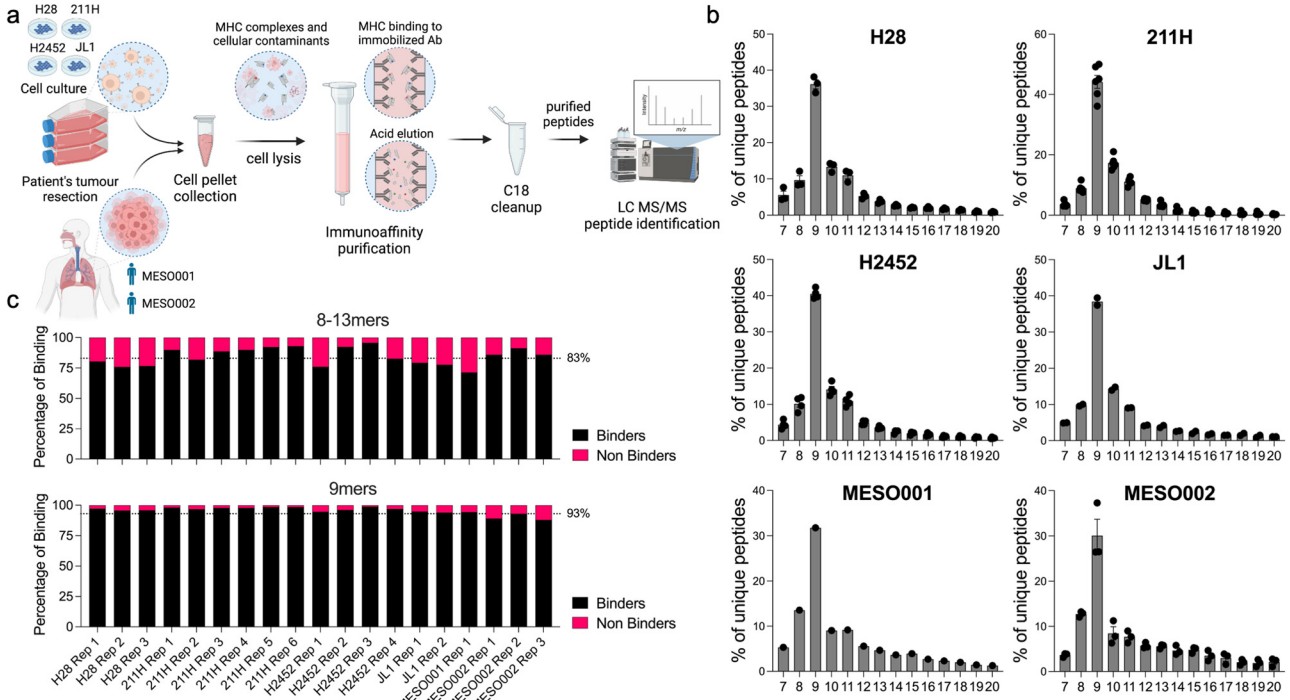

**Fig. 1 | Overview of the immunopeptidomics pipeline and quality control for eluted peptides. a** Infographic illustrating the immunopeptidomics pipeline employed to explore the MHC-I repertoire of human cell lines and resected tumors. The study included four human mesothelioma cell lines (H28, 211H, H2452, JL-1) and tumor resections from two different patients (MESO001, MESO002). Infographic was created with Biorender.com. **b** Distribution of peptide lengths as a percentage of the eluted peptides, revealing the expected peak at 9 amino acids across all analyzed samples. Bars represent the mean ± SEM. **c** Average percentage of

peptides predicted to be binders for each biological replicate, categorized by the interval of 8-13mers or specifically 9mers. The identification of "binders" was accomplished using the machine learning-based MHC-binding affinity tool MHCflurry, employing the corresponding set of MHCs for each sample. Peptides were classified as "binders" when their predicted "affinity percentile" was lower than 2. The number of independent samples for the data shown above was: H28 $n = 3$, 211H $n = 6$, H2452 $n = 4$, JL1 $n = 2$, MESO001 $n = 1$, MESO002 $n = 3$. Source data are provided as a Source Data file.

considering only the 9mers, the average percentage of MHC-specific peptides increased to 93% (Fig. 1c).

These results demonstrate the effectiveness of our technique in eluting peptides presented on the MHCs and confirm the high quality of our datasets.

To further investigate the quality of our immunopeptidomics datasets, we investigated the MHC-I binding motifs of the eluted peptides using an unsupervised clustering method (Gibbs-Clustering). We observed that the Gibbs-Clustering algorithm identified most of the expected binding motifs for all the expected MHC alleles. Missing identification of an expected binding motif could depend on an insufficient number of peptides per allele or shared key anchor positions between two or more alleles (Fig. 2a).

Conversely, peptide specificity assigned by the machine learning-based method included in PyptidOmicsQC (MHCflurry[17], based on "affinity Rank" score) provided a more intuitive representation and higher resolution of the HLA-binding specificity for complex haplotypes. It revealed that nearly all 9mers in each sample were predicted to be strong or weak binders for a specific MHC-I allele. This representation highlighted the unequal distribution of peptides across MHCs and intuitively showed the presence of some contaminant peptides in the MESO002 runs (Fig. 2b).

Next, we were interested in investigating how "shared" were the eluted peptides among samples, hence how "general" could our dataset be considered.

Therefore, we compared the sets of eluted peptides among different immunopeptidomics runs by calculating the percentage of pairwise overlap between the datasets of each peptide in each sample. We observed that cell lines mostly exhibited "private" epitope signatures, with minimal overlap between different cell lines (Fig. 3a).

Additionally, we explored whether this phenomenon could be explained by the similarity in MHC haplotype composition. Clustering the one-hot-encoded MHC alleles of different samples yielded the same pattern as before, suggesting that the previously observed clusters of closely related peptide profiles reflected the number of shared MHC alleles or superfamily of MHC alleles (Fig. 3b), once again indicating the significant involvement of MHC haplotype composition in comparing immunopeptidomics data.

Furthermore, we were interested in investigating the relationship between peptide binding affinity and source protein level in shaping the presented peptide repertoire.

We thus compared our dataset of eluted peptides with the quantitative proteomics datasets available at the Cancer Cell Line Encyclopedia (CCLE)[18]. Among the cell lines included in our study, MSTO-211H was the only one available in the CCLE proteomics dataset (https://gygi.hms.harvard.edu/publications/ccle.html). We observed that the average peptides intensity increased with the number of replicates in which certain peptides could be found (Supplementary Fig. 2A).

Next, we observed that, on average, the proteins presenting peptides on the MHC-I were significantly more abundant in MSTO-211H compared to proteins for which no MHC-I peptides were identified in our immunopeptidomics runs (Supplementary Fig. 2B). For the proteins for which we identified eluted peptides, we observed no correlation between the protein abundance and the number of times a given deriving peptide could be found (Supplementary Fig. 2C).

We also sought to investigate whether MHC-binding affinity alone could explain peptide presentation. Therefore, we in silico generated a set of 9mers by scanning all the source proteins of the peptides contained in our dataset and predicted their MHC-binding affinity using

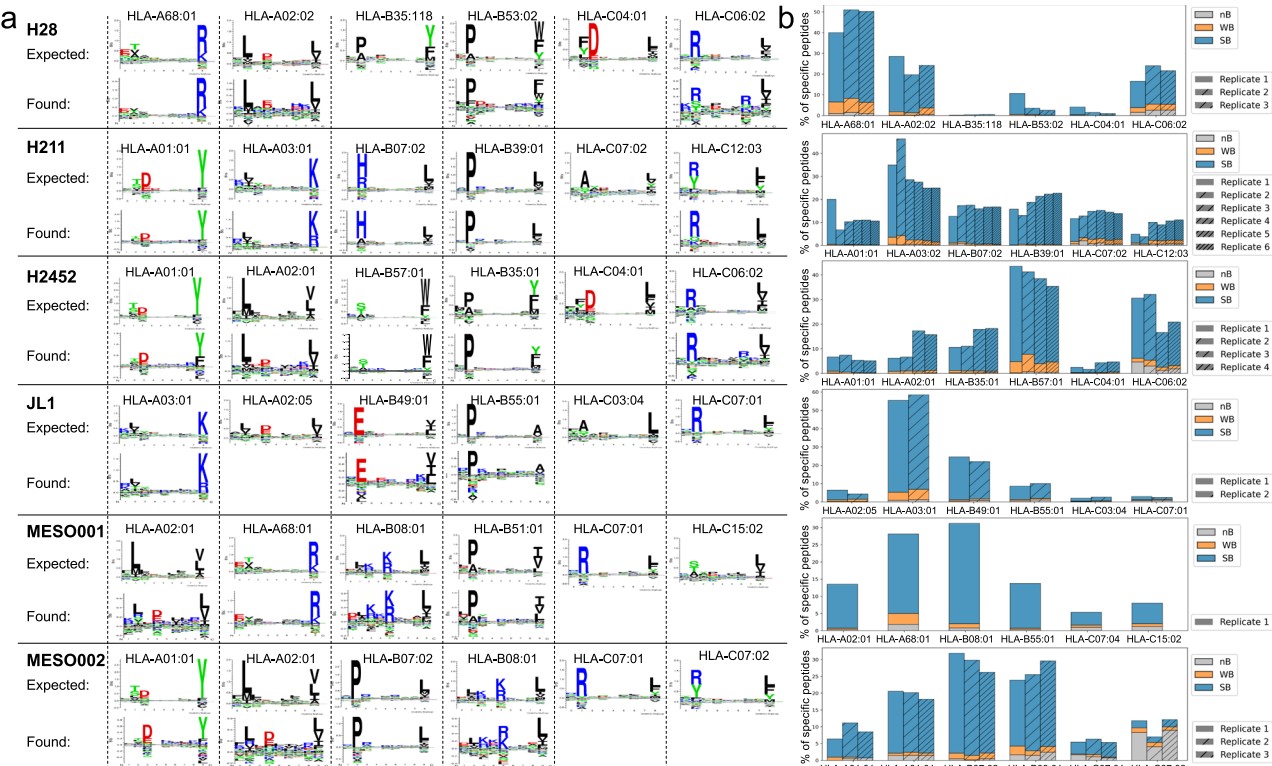

**Fig. 2 | Eluted peptides binding specificity deconvolution. a** Comparison between the expected motifs of specific MHC alleles obtained from NetMHCpan/Motifs viewer (Naturally presented ligands) (referred as "expected"), and the results of the unsupervised motif deconvolution method Gibbs clustering (referred as "found"). **b** Stacked bar plot illustrating the outcome of peptide-specificity deconvolution based on machine learning-based prediction performed using MHCflurry. Peptide specificity is annotated by using the best predicted peptide: MHC allele per each peptide. In the plot, the abbreviation "nB" represents "non-Binders," "WB" stands for "Weak Binders," and "SB" signifies "Strong Binders". The number of independent samples for the data shown above was: H28 $n = 3$, 211H $n = 6$, H2452 $n = 4$, JL1 $n = 2$, MESO001 $n = 1$, MESO002 $n = 3$. Source data are provided as a Source Data file.

MHCFlurry. We observed a weak correlation between peptide:MHC binding affinity and the number of times they were eluted. (Supplementary Fig. 2D).

Ultimately, we investigated the relationship between the peptide binding affinity and the abundance of its source protein. We found no obvious linear dependency, however, between the expression level of the source protein and the relative binding affinity of the corresponding presented peptides (Supplementary Fig. 2E).

Interestingly, a large number of peptides that were predicted to have a very high MHC binding affinity comparable to our eluted peptides were not identified in any of the immunopeptidomic runs. When considering only the "strong binders" (Rank ≤ 0.5) the dataset consisted of 139,634 predicted peptides, of which only 3325 were found in our immunopeptidomic runs (2.4%). This ultimately suggests that only a small fraction of strong-binding peptides have the appropriate characteristics, allowing them to be eventually presented on the cell surface.

Ultimately, we were interested in dissecting the quality and composition of our eluted peptides dataset in comparison to the previous knowledge in the field. To do that, we compared our set of eluted 9mers with the MHC ligands dataset deposited in IEDB (Immune Epitope Database and Analysis Resource) (www.iedb.org)[19]. We observed that an average of 86% of the peptides found in each run had already been reported in the literature (referred to as "Overall dataset") while 61% were found in the subset of peptides eluted from Healthy tissues (referred as "Healthy dataset"), and lastly 70.4% were found in the subset of peptides eluted by the remaining non-healthy conditions (referred as "non-healthy dataset") (Supplementary Fig. 3A).

A deeper analysis showed that already-known peptides had a general better predicted binding profile compared to those that had not been previously identified (Supplementary Fig. 3B).

Overall, our data suggested that most of our immunopeptidomic runs were of high quality and reproducible. However, we observed that the composition of the immunopeptidomic landscape heavily depends on the MHC haplotype composition affecting the comparison of the results obtained from samples of different sources. Additionally, we noticed that not all the MHC alleles foster the same number of peptides adding a further layer of complexity in shaping the immunopeptidome landscape. Lastly, we observed that a large portion of peptides eluted by our immunopeptidomics assays have already been previously annotated and deposited in public repositories conferring a final validation of the quality of our runs and proof that we were able to identify naturally presented MHC-I peptides.

**Peptides identified via immunopeptidomics show potential to be used as cancer antigens**

In order to demonstrate that MHC-I peptides identified through immunopeptidomics could be utilized for the development of therapeutic cancer vaccines, we assessed their ability to activate CD8+ T cells.

We first selected a pool of candidate peptides for immunological validation. We chose peptides with gene expression levels that were upregulated in mesothelioma, by using the datasets published by Morani et al.[13] and Barone et al.[20]. Among these, we initially prioritized the peptides eluted from the two resected tumor samples, and then we considered those that could be identified in other cell lines (the final list of selected peptides is shown in Supplementary Table 1). We prioritized peptides with highest MHC-binding affinity and ultimately, we favored peptides that were 9 amino acids in length over other lengths (Fig. 4a).

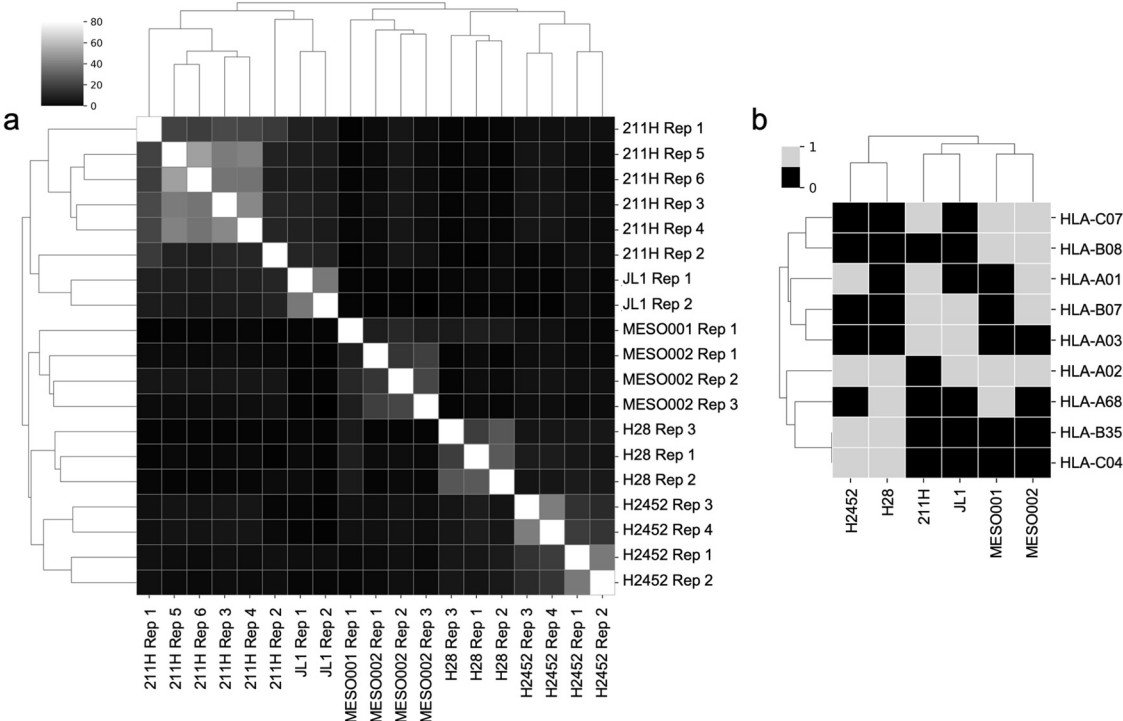

**Fig. 3 | Sample clustering based on immunopeptidomics-derived peptides and HLA haplotype composition. a** Hierarchical clustering of immunopeptidome overlap among different samples. The heatmap illustrates the pairwise overlap percentage between each sample. To enhance visualization, overlaps exceeding 40% are capped. **b** Clustering of samples, including cell lines and resected tumor samples, based on their HLA haplotype composition. The presence of alleles was represented using a one-hot encoding approach. A value of 1 indicates that a specific allele was "found" (represented by gray), while a value of 0 indicates that the allele was "not found" (represented by black). The number of independent samples for the data shown above was: H28 $n = 3$, 211H $n = 6$, H2452 $n = 4$, JL1 $n = 2$, MESO001 $n = 1$, MESO002 $n = 3$. Source data are provided as a Source Data file.

To assess the immunogenicity of the selected peptides, we initially obtained healthy donors' HLA-typed PBMCs from the Finnish Red-Cross Biobank service and we co-incubated them with our selected peptides. We observed that peptides 8 and 9 had the highest capacity of activating PBMCs compared to the other peptides (Fig. 4b and Supplementary Fig. 4A). Next, we tested our peptides in patients' PBMCs (MESO001, MESO002) and we observed that peptide 9 had a high ability to activate PBMCs (Fig. 4c and Supplementary Fig. 4B).

To assess whether some of the other peptides were immunogenic, we proceeded with co-incubating CD8+ T cells derived from healthy donors with autologous monocyte-derived dendritic cells (moDCs) pulsed with a mixture of the selected peptides (MIX-A and MIX-B) for 20–22 days to expand the population of peptide-specific T cells. Subsequently, we evaluated the specificity of CD8+ T cells in response to restimulation with single peptides, observing variable activation of CD8+ T cells upon restimulation with different peptides (Fig. 4d and Supplementary Fig. 4C). To assess whether the selected peptides were indeed of relevance for mesothelioma tumor cell lines, we performed the same protocol of CD8 T cell expansion using whole tumor lysates (from H28 and H2452 cell lines) and subsequently deconvoluted the CD8+ T cell specificity by restimulating these latter with the single peptides. In two different healthy donors we observed that the co-incubation of T cells with H28 tumor lysate resulted in the expansion of T cells specific for peptides 1, 6, 8, and 9. On the other hand, the co-incubation of donors' T cells with H2452 tumor lysate resulted in the expansion of T cells specific only for the peptides 1 and 6 (Fig. 4e and Supplementary Fig. 4D). Ultimately, we performed a killing assay using T cells expanded with our mix of peptides. Despite some unspecific killing was detected, we observed that T cells expanded with our mix of peptides showed higher killing capacity compared to T cells expanded

with a control peptide (MART1) in both the target cell lines H28 and H2452 (Fig. 4f).

Overall, our results demonstrated that some of the peptides identified through immunopeptidomics exhibited promising immunogenicity profiles, suggesting their potential for use in the development of therapeutic cancer vaccines for MPM.

## Peptides identified via immunopeptidomics can be used to develop therapeutic cancer vaccination

To test the robustness of our pipeline for formulating therapeutic vaccines using newly identified antigens, we performed immunopeptidomics analysis ($n = 4$ replicates) of a murine tumor model of malignant mesothelioma derived from mice exposed to asbestos fibers: AB12[21]. The AB12 cell line, which exhibits a biphasic phenotype[22], demonstrated high expression of MHC class I molecules on the cell surface, making it a suitable candidate for immunopeptidomics (Supplementary Fig. 5A). We performed immuno-affinity enrichment of MHC molecules, followed by peptide elution and LC-MS/MS, resulting in an average of 2000 unique peptides of varying lengths, with ~1500 peptides between 8-13 amino acids in length (corresponding to the 75% of all eluted peptides). Among them, around 1000 were unique 9mers (representing 50% of all eluted peptides) (Supplementary Fig. 5B). Additionally, the eluted peptides exhibited the expected length distribution, with an enrichment of 9-amino-acid peptides (Fig. 5a), and the motifs of the eluted peptides matched the expected binding motifs for MHCs H2-Kd and H2-Dd (Fig. 5b). Interestingly, peptide-specificity deconvolution consistently revealed a higher number of peptides belonging to H2-Kd compared to H2-Dd in all replicates (Fig. 5c).

To select suitable candidate peptides for formulating an AB12-specific therapy, we analyzed the gene expression levels of the genes corresponding to the eluted peptides. For this purpose, we examined

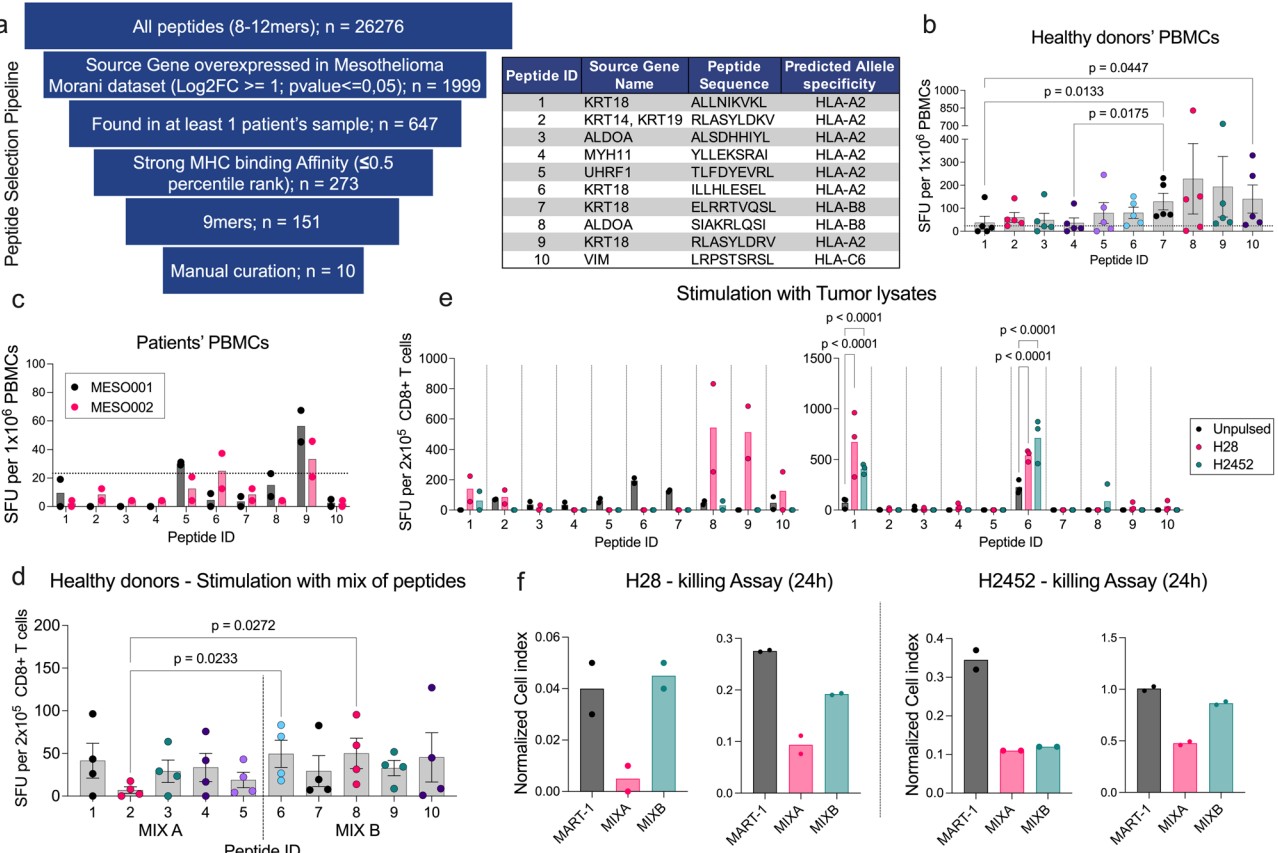

**Fig. 4 | Evaluation of in vitro immunogenicity for specific human MHC-I restricted epitopes. a** Illustration depicting the pipeline for peptide selection for immunogenicity assessment. The "manual data curation" entailed the application of supplementary criteria, which were flexibly applied to reach the final number of 10 peptides. Among these criteria, we considered the identification of peptides in one or more cell lines, or the presence of the source gene among the upregulated ones in mesothelioma, as documented in the study by Barone et al. **b** ELISpot assay measuring IFN-γ secretion in PBMCs obtained from $n = 5$ HLA-typed healthy donors upon stimulation with selected mesothelioma-derived peptides. Data are shown as bars plots representing the mean IFN-γ secretion ± SEM. **c** ELISpot readout displaying IFN-γ secretion in PBMCs derived from $n = 2$ cancer patients (MESO001, MESO002) upon stimulation with selected peptides. **d** ELISpot assay measuring IFN-γ secretion in purified T-cells obtained from $n = 4$ HLA-typed healthy donors. These T-cells were co-cultured with autologous moDC (monocyte-derived

dendritic cells) and different mixes of the selected peptides. Peptide Mix A includes peptides 1 to 5, while Peptide Mix B includes peptides 6 to 10. Bars represent the mean IFN-γ secretion ± SEM. **e** ELISpot readout of T-cells purified from $n = 2$ distinct HLA-typed healthy donors, which were co-cultured with their respective autologous moDCs pulsed with tumor lysate from either H28 and H2452 cell lines. T-cell activation specific to mesothelioma-derived epitopes was determined by restimulation with individual selected peptides. **f** Killing assay conducted using H28 or H2452 cell lines as target cells co-incubated with T-cells expanded with either mixes of mesothelioma-derived peptides (either MIX A or MIX B), or the control peptide MART-1. The killing assay data shown in the figure were the results of two independent experiments performed with T cells from two distinct donors. The statistical test used for all the panels was an ordinary one-way ANOVA with Fisher LSD test. Source data are provided as a Source Data file.

the gene expression dataset PRJEB15230 deposited in the EBI repository, where the authors investigated the alterations caused by asbestos exposure in mice. Malignant cells resulting from asbestos exposure exhibited different gene expression profiles compared to healthy pleura and exposed pleura that did not develop a tumor (Supplementary Fig. 6A). We initially focused on peptides derived from proteins that were overexpressed in the tumor compared to healthy pleura (Supplementary Fig. 6B). Furthermore, we considered binding affinity, fold change of expression in the tumor compared to healthy tissue, and the number of replicates in which each peptide was identified. Specifically, we selected peptides with a predicted binding affinity below 50 nM, a log fold change >2, and found in three or more replicates. Based on these criteria, we selected a pool of 15 candidate peptides (Fig. 5d and Supplementary Table 2).

Next, we assessed the immunogenicity of the selected peptides by immunizing cohorts of mice with different vaccination schedules (Fig. 5e, f), subcutaneously injecting a mixture of short-synthesized peptides and poly I:C as an adjuvant. Spleen cells from immunized mice were collected at the endpoint and used for ELISpot analysis.

The immunogenicity assessment revealed only partial overlap, with peptides 2 and 7 exhibiting T cell immunogenicity only after the short vaccination regimen. Peptides 5, 11, and 12 displayed more consistent immunogenicity and elicited higher responses upon splenocyte restimulation, both after the short and long vaccination schedules (Fig. 5g, h and Supplementary Fig. 7A, B). Additionally, we observed that splenocytes from immunized mice were able to kill AB12 cells (Fig. 5i).

To further confirm the relationship between the AB12 tumor model and the identified immunogenic epitopes, we sought to determine which epitopes naturally elicited an immune response when mice were exposed to the tumor. Mice were immunized using AB12 tumor lysate and an oncolytic adenovirus encoding mOX40L-mCD40L (VALO-mD901) as an adjuvant, following the schedule shown in Fig. 5j. ELISpot analysis conducted after immunization revealed that all peptides induced a variable release of interferon-γ upon restimulation of splenocytes in mice injected with the combination of tumor lysate + VALO-mD901. Peptides 11 and 12 were shown to be the most immunogenic candidates (Fig. 5k and Supplementary Fig. 7C).

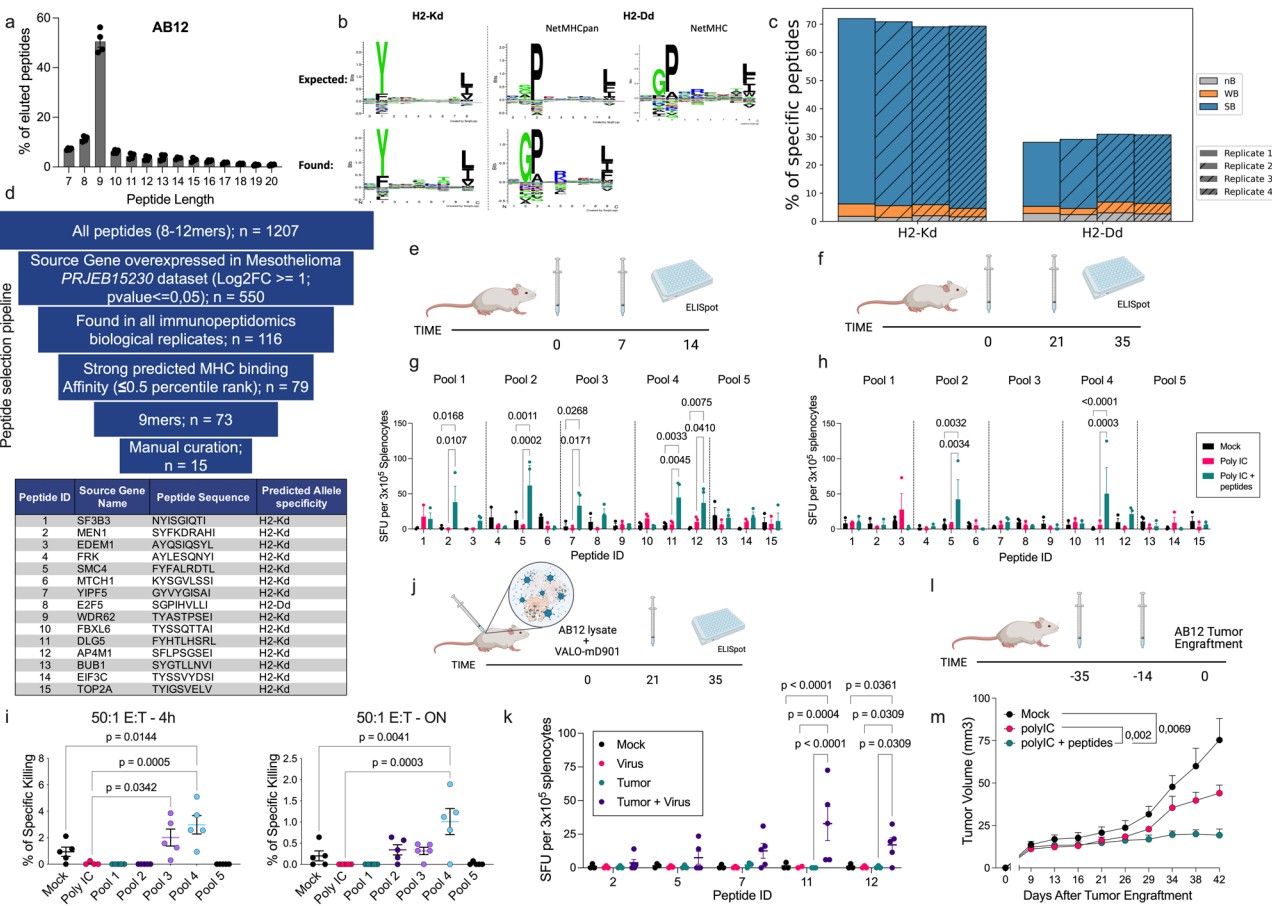

**Fig. 5 | Characterization of immunopeptidomic landscape of the AB12 cells and immunological validation of selected peptides. a** Eluted peptides length distribution. Bars show mean of $n = 4$ biological replicates ±SEM. **b** Comparison between the expected Balb-c MHCs binding motif ("expected") with the one obtained from the eluted peptides ("found"). **c** Stacked bar plot representing peptide-specificity deconvolution (predicted by MHCflurry) for the $n = 4$ independent replicates. **d** Schematic illustrating the peptide selection pipeline for subsequent immunogenicity screening. As "Manual Curation", we considered the process of selection of the 15 candidate peptides which best fulfilled the reported selection criteria. **e** Schematic of the short immunization protocol. **f** Schematic of the long immunization protocol. **g** ELISpot readout of the short immunization protocol. Bars show mean Spot Forming Units (SFUs) of $n = 3$ biological replicates ±SEM. **h** ELISpot readout of the long immunization protocol. Bars show mean SFUs of $n = 3$ biological replicates ±SEM. **i** LDH Killing assay performed by coincubating

AB12 cells with splenocytes derived from immunized mice for 4 or 16 h (ON). Graphs show mean specific killing percentage of $n = 5$ biological replicates ±SEM. **j** Schematic of the immunization protocol using AB12 cell lysate. Mice were subcutaneously injected with PBS (Mock, $n = 3$), VALO-mD901 (Adenovirus) (Virus Alone, $n = 3$), or AB12 cells lysate (Tumor Lysate, $n = 3$), or AB12 cells lysate + VALO-mD901 (Adenovirus) (Tumor Lysate + Virus, $n = 5$). **k** ELISpot readout of the immunization protocol with tumor lysate. Bars show mean Spot Forming Units (SFUs) ± SEM of 3 or 5 biological replicates. **l** Schematic of the immunization regimen followed by AB12 tumor engraftment. **m** Average tumor growth curve depicted as mean (+SEM) for each immunization group ($n = 10$ mice per group). Statistical significance for the data shown in (**g, h, k, m**) was evaluated using two-way ANOVA with Fisher LSD test. While for the (**i**), statistical significance was evaluated using one-way ANOVA with Tukey correction. All the schematics were created using Biorender.com. Source data are provided as a Source Data file.

Lastly, to confirm that immune response toward peptides 11 and 12 could impact tumor progression in vivo, we immunized the mice with the peptides followed by AB12 tumor engraftment (Fig. 5l) and we observed that indeed the tumor growth was indeed reduced in immunized animals (Fig. 5m).

Next, we aimed to assess the efficacy of peptide-specific therapeutic cancer vaccination. For this purpose, we selected the two peptides (11 and 12) with the best immunogenicity profiles identified in our screening. We employed PeptiCRAd technology, which we had previously developed in our laboratory[6], as a delivery system for the selected peptides. PeptiCRAd is a vaccine platform comprising an oncolytic adenovirus (OAd) whose capsid is decorated, via electrostatic interaction, with tumor-derived peptides elongated at the N-terminus with a poly-lysine tail (poly-K peptides). This platform has demonstrated success in generating multiple proofs-of-concept for the treatment of various murine tumor models[7–11]. To produce PeptiCRAd, a poly-K version of the previously selected peptides was synthesized. We then assessed the interaction between poly-K peptides

and the viral capsid using surface plasmon resonance (SPR). Our data indicated a weak interaction between Poly-K-tailed peptides and the negatively charged capsid of Adenovirus (Fig. 6a).

Mice were preimmunized using PeptiCRAd (VALO-mD901 coated with peptides 11 and 12) according to the schedule depicted in Fig. 6b. Subsequently, spleens were collected and splenocytes were co-cultured on AB12-eGFP/Luc to evaluate their tumor killing capacity. Our results show that splenocytes derived from immunized animals showed significantly higher killing compared to either Mock or Virus Alone (uncoated VALO-m901) controls (Fig. 6c).

To evaluate the efficacy of PeptiCRAd on established tumors, we subcutaneously engrafted a cohort of mice with AB12 tumor cells. When the tumors had established (21 days after tumor implantation), the mice were intratumorally treated with PeptiCRAd (Ad5/3-D24 coated with peptides 11 and 12) (Fig. 6d). Mice treated with PeptiCRAd exhibited enhanced control of tumor growth compared to those treated with PBS alone (Mock) or adenovirus alone (Uncoated Ad5/3-D24) (Fig. 6e). Moreover, the therapeutic success rate increased from

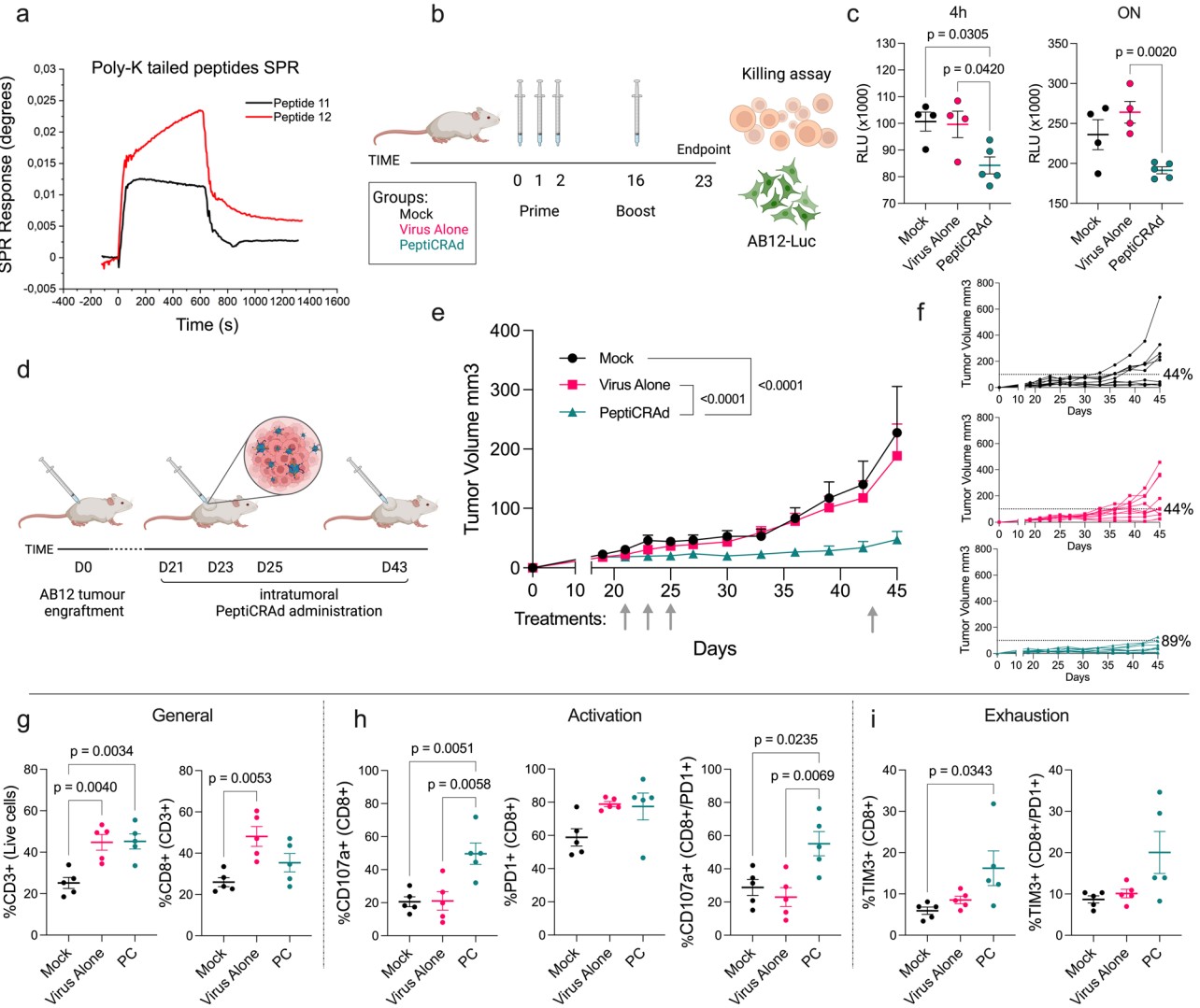

**Fig. 6 | Therapeutic vaccine utilizing PeptiCRAd technology for the treatment of established AB12 tumors. a** Surface plasmon resonance (SPR) response of selected peptides interacting with the virus capsid surface. **b** Schematic representation of the immunization regimen for the killing assay. **c** Killing assay results presented as mean relative light units (RLU) emitted by live Luc + AB12 incubated with luciferin ± SEM (Mock n = 4, Virus Alone n = 3, PeptiCRAd n = 5 biological independent replicates). Luc + AB12 cells were co-incubated with splenocytes derived from immunized animals at a 50:1 Effector-to-Target (E:T) ratio for either 4 h or 16 h (ON). Statistical significance was assessed using one-way ANOVA and Tukey's correction. **d** Experimental layout for the treatment of established AB12 tumors. **e** Average tumor growth curve depicted as mean (+SEM) for each treatment group (n = 9 mice per group). Statistical significance was evaluated using

two-way ANOVA. **f** Tumor volume curves of individual mice in each treatment group. The dotted line represents the threshold of therapeutic success, determined by the median tumor volume of the Mock group on the final day. **g** Immunological profile of tumor-infiltrating lymphocytes (TILs). This includes the frequency of the intratumoral CD3+ population and the frequency of CD8+ cells. **h** Frequency of CD107a+, PD1+, and CD107a+/PD1+ double-positive CD8 T cells. **i** Frequency of TIM3+ and TIM3+/PD1+ double-positive CD8 T cells. All data are presented as dot plots showing the mean population frequency percentage ± SEM and each dot represent a single mouse in each treatment group (n = 5). Statistical significance was assessed using one-way ANOVA and Tukey's correction. Schematics for experiments were created using Biorender.com. Source data are provided as a Source Data file.

44.4% for the Mock and Virus Alone groups to 89% for the PeptiCRAd-treated animals (Fig. 6f).

Tumors were harvested for downstream flow cytometric analysis to investigate the immunological modulation induced by the treatments (Supplementary Fig. 8). Interestingly, our analysis revealed differences in the frequency of infiltrating CD3+ T cells in tumors treated with PeptiCRAd or Virus Alone compared to the Mock group. However, despite effectively controlling tumor growth, PeptiCRAd-treated mice showed a lower infiltration of CD8+ T cells compared to those treated with Virus Alone (although not significant) (Fig. 6g).

Consistent with their tumor control capability, PeptiCRAd-treated mice exhibited a significantly higher intratumoral frequency of CD107a + positive CD8 T cells compared to the controls. However, although we only observed only a trend of a higher frequency of PD1+ tumor-

infiltrating CD8 T cells in mice treated with virus alone or PeptiCRAd, the latter showed a significantly higher frequency of double-positive PD1+CD107a+ cells compared to the controls (Fig. 6h). Within the tumor, we observed a tendency of slightly higher numbers of TIM3+ CD8 T cells or double-positive PD1+TIM3+ CD8 T cells in mice treated with PeptiCRAd (Fig. 6i). We also assessed the immune modulation caused by the different treatments in secondary lymphoid tissues (spleens). Interestingly, we observed no statistically significant differences in the frequency of either CD3+ or CD8+ T cells among the treatment groups (Supplementary Fig. 9A). However, in contrast to the tumor, we observed a similar enrichment of CD107a+ positive CD8 T cells in both mice treated with PeptiCRAd and adenovirus alone. Moreover, we observed no differences in the frequency of PD1+ T cells but noted a trend toward more PD1+/CD107a+ double-positive cells in

the PeptiCRAd-treated mice (Supplementary Fig. 9B). Finally, we observed a statistically significant increase in TIM3+ T cells in PeptiCRAd-treated mice compared to the controls, along with a more pronounced PD1+/TIM3+ double-positive T cell population in PeptiCRAd-treated mice (Supplementary Fig. 9C), aligning with the observations made in the tumor milieu.

Overall, our data demonstrate that immunogenic peptides identified via immunopeptidomics are tumor-specific and can be used to impact in vivo tumor progression. Indeed, the intratumoral administration of PeptiCRAd coated with AB12-derived immunogenic peptides induced a potent anti-tumor immune response and produced AB12 tumor growth control in tumor-bearing mice. These findings suggest that immunopeptidomics can be harnessed to generate effective therapeutic cancer vaccines for mesothelioma.

## Discussion

In the current study, we have explored the landscape of peptides presented on the tumor cell surface (immunopeptidome), aiming to reveal immunogenic T cell epitopes to use as possible targets for the development of a therapeutic cancer vaccine.

We observed that the immunopeptidome is heavily dependent on the MHC haplotype. Indeed, upon the comparison of eluted peptides from different source materials (samples from different subjects carrying different HLA haplotypes), we realized that the overlap increased together with the number of shared MHCs or binding motifs. These observations were also confirmed by Marcu et al.[23] who employed immunopeptidomics to analyze samples from several healthy tissues of different donors. The authors reported a higher similarity between immunopeptidomes and the composition of source proteins in samples originating from the same subjects compared to samples originated from the same tissue types but from different subjects[23]. This might suggest that MHC haplotype composition might have a more critical role in shaping the repertoire of presented epitopes compared to gene expression or protein content at the cellular or tissue level.

We found that highly expressed proteins were more likely to gather presented peptides[24,25], although it has been reported that this is not always the case[26]. Moreover, we observed that presented peptides have a generally very high affinity for the MHC. Although, beyond affinity, other factors such as (1) stability of the peptide in the MHC cleft and (2) turnover of the source protein; (3) MHC peptide cleavage/processing pathways should also be taken into consideration.

Interestingly, a substantial portion of the immunopeptidome contains peptides already annotated and deposited in public repositories, especially the peptides with strong binding affinity.

When we compared our dataset of naturally eluted peptides from MSTO-211H with an in-silico dataset, which was generated using the MHC-binding affinity prediction tool on the list of 9mers generated from source proteins found in the quantitative proteomics data deposited in the CCLE, only the 2.4% of the high affinity predicted peptides could be identified in our immunopeptidomics run. However, as also noted by Bassani-Sternberg et al.[24], not all predicted peptides are presented on the surface of living cells, highlighting the high number of false-positive peptides derived by prediction tools and simultaneously that the immunopeptidomics analysis is only a snapshot of what could be presented on the cell surface at a given time. Far more replicates would be needed to explore the whole MHC-I ligandome of a tumor.

Although mesothelioma is not generally considered a "hot" tumor, it is still infiltrated by CD8+ cytotoxic T cells, and, for this reason, from an immunological point of view, it is defined as "altered"[27]. In this context, active immunotherapies such as cancer vaccines could offer a promising solution; however, possible targets (such as tumor-associated antigens—TAA or tumor-specific antigens—TSA) for MPM are still largely unknown and poorly explored[14].

Two prominent candidates have been explored for immunotherapies in MPM as they appear to be upregulated in several tumors[28]: mesothelin (*MSLN*) and *WT1*. However, in our datasets of 12 human samples, we found only 1 eluted peptide derived from mesothelin, RVRELAVAL, also previously annotated by Nicholas et al.[29]. On the other hand, no peptides derived from the *WT1* protein were found in our dataset. This could be either due to the lack of sufficient replicates to capture the whole immunopeptidome landscape or that these proteins are only a minor source of CD8+ T cell epitopes, at least for the MHCs reported in the current study.

Nevertheless, in this study, we have identified several candidate target epitopes (listed in Supplementary Table 1), that could have potential use for the development of cancer vaccines Indeed, we have identified peptides derived by *KRT18*, *ALDO*, *UHRF1* and *VIM* that are known unfavorable prognostic markers for other cancers (source: the human protein atlas). Moreover, *UHRF1*, which is an epigenetic driver of mesothelioma, has previously been observed as a potential druggable target[30]; we found the presence of proteins related to extracellular matrix (ECM), such as cytokeratins which are particularly useful in the diagnosis of MPM, since all mesotheliomas potentially show high expression[31–33], and *VIM* which is highly expressed in up to 75% of malignant mesotheliomas[34]. Interestingly, *MYH11*, which was included in our set of genes overexpressed in mesothelioma, was observed to have an even higher expression in mesothelial hyperplasia[35]. Lastly, we found *ALDOA*, which has been observed to be a key enzyme involved in lung cancer[36].

In the second part of the study, we focused on providing a proof-of-concept for the use of immunopeptidomics for the development of vaccines against MPM in vivo.

For the development of cancer vaccines, selecting the suitable adjuvant and the schedule/regime of vaccine administration is as important as selecting the right target[26].

In this context, we have previously demonstrated that PeptiCRAd technology represents a versatile and effective platform for the development of cancer vaccines. Oncolytic viruses, beyond direct cytolysis of cancerous cells, stimulate the immune system toward the formation of a more immunogenic tumor microenvironment[3], promoting an anti-tumor immune response. Specifically, PeptiCRAd combines the immunogenicity of the adenovirus with the tumor specificity conferred by the peptides, eliciting a strong anti-tumor immune response. In this study, we showed that PeptiCRAd technology was successfully used to treat the established mesothelioma tumor model in vivo, providing a proof-of-concept that peptides identified by immunopeptidomics can be used for the development of cancer vaccines, as we have also previously demonstrated for other tumor models[8,10].

Current literature in the field of therapeutic cancer vaccines focuses mainly on identifying and validating tumor-specific antigens (TSAs) or neoepitopes, which arise from mutated proteins[37,38]. However, MPM has a low mutation burden;[16] for this reason, other routes, such as the use of tumor-associated antigens (TAAs), need to be favored. The latter represents an appealing possibility, as TAAs might be shared among several tumor types and different patients, but as "being self" they occasionally have poorer immunogenicity. Conversely, TSAs are more immunogenic, as they might escape the central immune tolerance more easily, but they tend to be private and "patient-specific"[26,39].

When selecting TAA-derived epitopes, one common criterion is represented by the assessment of differential gene expression between the tumor and the relative healthy tissue. However, despite the RNA-sequencing data of mesothelioma samples and RNA-array deposited in TCGA and other public repositories, it is difficult to find recent Omics of healthy mesothelium samples for a more fair and accurate differential gene expression analysis. Interestingly, in the work of Morani et al.[13] (which was used in the current study),

differential expression analysis was performed using healthy lung samples as a reference.

The analysis of the resected tumor was limited by the scarce biological material available for the analysis, making it impossible to obtain a sufficient number of replicates for immunopeptidome analysis. In the future, other solutions for the performance of such techniques could be considered, for example, an innovative microscale MHC affinity purification system, PeptiCHIP, which is ideal for small sample sizes[40].

In conclusion, we observed that the composition of the immunopeptidome landscape heavily depends on the MHC haplotype composition and that not all the MHC alleles foster the same number of peptides. Interestingly, we observed that most peptides eluted by immunopeptidomics in this study had been previously identified in other tumor types and deposited in public repositories, while they corresponded to just a fraction of the ones predicted by state-of-the-art machine learning-based tools. We validated a set of peptides eluted from human mesotheliomas and observed that they were able to promote T cell killing of mesothelioma cell lines. Lastly, we offer a proof-of-concept in mice that immunogenic peptides identified via immunopeptidomics can effectively be used to generate cancer vaccines.

## Methods

### Ethical permits

Patient's samples were received by the Helsinki University Hospital under the approval of the ethical review board (review number HUS/970/2021) and Helsinki University Hospital Institutional review board (IRB) (approval HUS/8/2022).

All animal experiments were reviewed and approved by the Experimental Animal Committee of the University of Helsinki and the Provincial Government of Southern Finland (license numbers ESAVI/11895/2019 and ESAVI/12722/2022). The maximal tumor size/burden allowed by our ethical permit is 18 mm in diameter. The maximal tumor size was never exceeded in any of the experiments carried in the current study.

### Cell lines and reagents

Human mesothelioma cell lines NCI-H2452, NCI-H28, and MSTO-211H were cultured in RPMI 1640 supplemented with 10% fetal bovine serum (FBS) (Gibco), 1% GlutaMAX (Gibco), and 1% penicillin-streptomycin (10,000 U/ml) (Gibco). The Finnish Institute for Molecular Medicine (FIMM, Helsinki) kindly donated all the above-mentioned human mesothelioma cell lines.

Human mesothelioma cell line JL1 was kindly donated by Dr. Kuryk (Valo Therapeutics Oy, Helsinki, Finland) and was cultured in DMEM supplemented with 20% fetal bovine serum (FBS) (Gibco), 1% GlutaMAX (Gibco), and 1% penicillin-streptomycin (10,000 U/ml) (Gibco). In addition, A549 cell line was obtained by ATCC and was cultured using DMEM low glucose supplemented with 10% fetal bovine serum (FBS) (Gibco), 1% GlutaMAX (Gibco), and 1% penicillin-streptomycin (10,000 U/ml) (Gibco).

Murine mesothelioma cell line AB12 was kindly donated by Dr. Kuryk and was cultured in RPMI 1640 supplemented with 5% fetal bovine serum (FBS) (Gibco), 1% GlutaMAX (Gibco), and 1% penicillin-streptomycin (10,000 U/ml) (Gibco). All cells were cultured at 37 °C, 5% $CO_2$ in a humidified incubator.

### Immuno-affinity purification of MHC class I peptides

MHC class I peptides were immunoaffinity purified from the AB12 mouse cell line (4 replicates) using inVivoMAb anti-mouse MHC Class I (H-2K$^d$, H-2D$^d$) (clone 34-1-2S, BioXCell, BE0180, Lebanon, NH, USA). In addition, MHC class I peptides were immunoaffinity purified from the H2452 ($n = 4$), H28 ($n = 3$), MSTO-211H ($n = 6$), and JL1 ($n = 2$) human cell line and from two patients' samples (1 replicate for MESO001 and three

replicates for MESO002, of which one from benign and two from tumor tissues) using anti-human HLA-A, HLA-B, and HLA-C antibodies (inVivoMAb, clone W6/32, BioXCell, BE0180, Lebanon, NH, USA). For sample preparation, the snap-frozen cell pellet ($1 \times 10^8$ cells for each replicate) was incubated for 2 h at 4 °C in lysis buffer. The lysis buffer contained 150 mM NaCl, 50 mM Tris-HCl, pH 7.4, protease inhibitors (A32955, Thermo Scientific Pierce, Waltham, MA), and 1% Igepal (I8896, Sigma-Aldrich, St. Louis, MO). The lysates were precleared by low-speed centrifugation for 10 min at $500 \times g$, and then the supernatant was centrifuged for 30 min at $25,000 \times g$. The cleared lysate was loaded to the immunoaffinity column (AminoLink Plus Immobilization, Thermo-Fischer) where 2 ml of pre-packed Agarose Resin were covalently linked to 1 mg of W6/32 antibody (inVivoMAb, BioXCell) via the linking procedure at neutral pH (pH = 7.2) following the manufacturer's instructions. Once binding occurred, the affinity column was washed using the following buffers: 150 mM NaCl, 20 mM Tris-HCl; 400 mM NaCl, 20 mM Tris-HCl; 150 mM NaCl, 20 mM Tris-HCl, and 20 mM Tris-HCl, pH 8.0; and bound complexes were eluted in 0.1 M acetic acid. Eluted peptides and the subunits of the MHC complexes were desalted using SepPac-C18 cartridges (Waters, WAT054960). The cartridge was prewashed with 80% acetonitrile in 0.1% trifluoroacetic acid (TFA) and then with 0.1% TFA. The peptides were purified from the MHC class I protein chains by elution with 30% acetonitrile in 0.1% TFA. Finally, the samples were dried using vacuum centrifugation (Eppendorf).

### LC-MS analysis of MHC class I peptides

Each dry sample was dissolved in 10 µl of LC–MS solvent A (0.1% formic acid) by dispensing/aspirating 20 times with the micropipette. The nanoElute LC system (Bruker, Bremen, Germany) injected and loaded the 10 µl of sample directly onto the analytical column (Aurora C18, 25 cm long, 75 µm i.d., 1.6 µm bead size, Ionopticks, Melbourne, Australia) constantly kept at 50 °C by a heating oven (PRSO-V2 oven, Sonation, Biberach, Germany). After washing and loading the sample at a constant pressure of 800 bar, the LC system started a 30 min gradient from 0 to 32% solvent B (acetonitrile, 0.1% formic acid), followed by an increase to 95% B in 5 min, and finally a wash of 10 min at 95% B, all at a flow rate of 400 nl/min. Online LC-MS was performed using a Tims TOF Pro mass spectrometer (Bruker, Bremen, Germany) with the CaptiveSpray source, capillary voltage 1500 V, dry gas flow of 3 l/min, and dry gas temperature at 180 °C. MS data reduction was enabled. Mass spectra peak detection maximum intensity was set to 10. Mobilogram peak detection intensity threshold was set to 5000. Mass range was 300–1100 $m/z$, and mobility range was 0.6–1.30 V s/cm². MS/MS was used with 3 PASEF (parallel accumulation–serial fragmentation) scans (300 ms each) per cycle with a target intensity of 20,000 and an intensity threshold of 1000, considering charge states 0–5. Active exclusion was used with release after 0.4 min, reconsidering a precursor if the current intensity is >4-fold the previous intensity, and using a mass width of 0.015 $m/z$, and a $1/k_0$ width of 0.015 V s/cm². Isolation width was defined as 2.00 $m/z$ for mass 700 $m/z$ and 3.00 $m/z$ for mass 800 $m/z$. The collision energy was set as 10.62 eV for $1/k_0$ 0.60 V s/cm² and 51.46 eV for $1/k_0$ 1.30 V s/cm². Precursor ions were selected using 1 MS repetition and a cycle overlap of 1 with the default intensities/repetitions schedule.

### Proteomics database search

All MS/MS spectra were searched by PEAKS Studio X+ (v10.5 build 20191016) using a target–decoy strategy. The database used was the Swissprot Human protein database (including isoforms, 42,373 entries, downloaded from uniprot.org on 2019-11-26).

A precursor mass error tolerance of 20 ppm and a fragment mass error tolerance of 0.02 Da were used. The Enzyme was "None", digest mode was "Unspecific", and oxidation of methionine was used as variable modification, with a maximum of three oxidations per

peptide. A false discovery rate (FDR) cutoff of 1% was employed at the peptide level. The mass spectrometry proteomics data have been deposited to the ProteomeXchange Consortium via the PRIDE partner repository with the data set identifier PXD038273.

## MESO002 immunopeptidome data cleanup

The data obtained from MESO002 samples revealed the presence of contaminant peptides with longer lengths. These peptides exhibited a characteristic "ladder profile," suggesting that they were proteolytic contaminants. In order to detect such contaminants and flag contaminating source proteins, we defined a *protein coverage ratio* for each protein (P) as average number of class I peptides (p) per amino acid, as shown in the work of Fritsche et al.[41]:

$$\text{protein coverage ratio for protein P} = \frac{1}{L(P)} \times \sum_{p \in P} L(p); \quad (1)$$

$L(x)$ = Number of amino acids per peptide (p) or its source protein (P).

To identify a suitable cutoff for the identification of contaminant peptides, were calculated these scores using as reference the data derived from "clean" runs (e.g., the immunopeptidomics run of the mesothelioma cell lines) and set the cut-off at the average 95th percentile of those distribution.

## In silico analysis of MHC-class I peptides

The immunopeptidomics quality check was conducted using an in-house script called "PyptidOmicsQC" which is publicly available on GitHub (URL will be provided upon acceptance of this manuscript). PyptidOmicsQC is a web application developed in Flask, offering a user-friendly interface, and facilitating quick visualization of QC graphs through a dashboard. The script was written in Python, specifically using version 3.9.7 (from Anaconda version 2021.11), and it operates entirely within a Python environment.

Briefly, this tool automatically provides (1) peptide length distribution (in raw numbers and percentage) of all the provided replicates at once side by side. (2) Representation of overlap between replicates is performed using the UpSet plot (from Upsetplot package: https://upsetplot.readthedocs.io/en/stable/), run with default settings. (3) Stacked bar plot deconvoluting the eluted peptides' allele specificity. Peptide specificity was assessed by MHCflurry package using allele rank for the best allele predicted. Peptides ranking below 0.5% were considered "strong binders", while peptides ranking between 0.5% and 2% were considered "weak binders". Peptides scoring above 2% are considered "non-binders". MHCflurry was favored over other viable options not only as it showed to have a better performance in peptide MHC-binding affinity prediction[17], but also because MHCflurry is an open-source software coded in Python and available as a PyPI package. This feature made it easier to integrate in a Python environment such as PyptidOmicsQC.

Throughout the study, MHC-binding affinity was predicted using either MHCflurry 2.0[17] or NetMHCpan 4.1[42], with the same threshold as described above.

Unsupervised clustering analysis of peptides into groups based on sequence similarities was performed using the GibbsCluster-2.0 tool with the default setting.

The known MHC motifs were obtained from the Motif Viewer section of either NetMHCpan 4.1 or NetMHC (DTU Bioinformatics).

## One-Hot-Encoding

One-Hot-Encoding is a common technique employed in machine learning to represent categorical variables as binary vectors, converting them into a numerical format. It involves constructing a unique binary vector for each categorical variable. In this vector representation, only the element corresponding to the categorical value is set to 1 ("hot"), while all other elements are set to 0 ("cold").

One-Hot-Encoding was performed to allow clustering of the MHC haplotypes of the samples considered in the study. Only the HLA alleles shared by more than one sample were considered for building the binary vector. Four digits alleles names (e.g., HLA-A02:01) were reduced to the first 2 digits (A02), as long as all the alleles aggregated in this way presented the same peptide-binding motif.

## Production of predicted presented peptide dataset of MSTO-211H

The complete amino acid sequences of the proteins identified through quantitative mass spectrometry of the MSTO-211H, available in the Cancer Cell Line Encyclopedia CCLE[18], were obtained from the canonical reviewed human proteome (UP000005640). Each full-length protein sequence that had at least one eluted peptide in the dataset, was fed into MHCflurry. A scan of the whole protein sequences was performed considering only 9 amino acid long peptides for the output. The MHC-binding affinity was computed for all the MHC alleles composing the haplotype of the MSTO-211H (http://celllines.tron-mainz.de/).

## Adenoviruses preparation

For the animal experiments, we used the viruses Ad5/3-D24 and VALO-mD901[43]. The Ad5/3-D24 is a conditionally replicating adenovirus of the chimeric 5/3 serotype with a 24-base pair deletion in the E1A region (Ad5/3-D24). VALO-mD901 has the same backbone but, additionally, the Adenoviral E3 region was replaced by the expression cassette constituted by the human CMV promoter, followed by murine OX40L and murine CD40L genes separated by a 2A self-cleaving peptide sequence, and finally β-rabbit globin polyadenylation (ploy A) signal. VALO-mD901 was previously generated using a methodology shown elsewhere[44]. Both Ad5/3-D24 and VALO-mD901 were amplified using A549 cells. After amplification, both viruses were purified using double-cesium chloride gradients and stored at −80 °C in an A195 adenoviral storage buffer. The viral particle (vp) concentration was measured at 260 nm.

## Peptides

The short and poly-K murine peptides were purchased from Chempeptide limited (Shanghai, China), while the short and long human peptides were purchased from GenScript (USA). For additional information, see Supplementary Table 3.

## Mesothelioma tumor samples

The mesothelioma tumor biopsies and blood were collected from two patients who underwent surgical removal of the tumors. Tumor samples were collected and delivered directly from HUS Hospital and patients gave their written consent. Sex or gender were not considered relevant in the current study.

Cells were isolated from the original tissue after surgery by physically mincing the resected tumor into small pieces using a scalpel; subsequently, the tissue was treated using human tumor dissociation buffer (Miltenyi Biotech) following manufacturer instructions and the Gentle MACS dissociator (using the default human tumor program).

## HLA typing of patient's tumor samples

The targeted PCR-based next-generation sequence (NGS) technique was used to perform allele determination of three classical HLA-I genes, HLA-A, HLA-B, and HLA-C, according to the protocol provided by the manufacturer (NGSgo Workflow, GenDx, Utrecht, The Netherlands). The allele assignment at the four-field resolution level was implemented by NGSengine version 2.21.0.20156 (GenDx, Utrecht, The Netherlands) using IPD IMGT/HLA database, release 3.43.0; https://www.ebi.ac.uk/ipd/imgt/hla/.

## Human healthy donors and patient's PBMCs collection

HLA-matched fresh buffy coat products were obtained from the Blood Service Biobank of the Finnish Red Cross Blood Service. On the same day, the HLA types of the donated blood units from donors who provided valid biobank consent were retrieved from the Finnish Red Cross database using a custom-built script. As part of the standard production process, the buffy coat layer of blood units carrying the desired HLAs was separated as part of the regular production process. After pseudonymization the buffy coat products were handed over for research purposes.

Cancer patients' full blood was collected at the same time as the surgery. SepMate separation columns (StemCell Technologies, cat:85450) were used to isolate PBMCs from buffy coats according to manufacturer instructions. The PBMCs were subsequently cultured in RPMI 1640 supplemented with 5% human AB serum (Capricorn), 1% GlutaMAX (Gibco) and 1% penicillin-streptomycin (10,000 U/ml) (Gibco), 1% MEM Non-Essential Amino Acids (NEAA) (Sigma), Sodium pyruvate 1 mM (Gibco).

The list of PBMCs or buffy coats samples and the corresponding HLA typing are provided in Supplementary Table 4.

## Generation of peptide-specific T cells

Anti-CD14 microbeads (Miltenyi Biotec) were used to isolate monocytes from PBMCs, and anti-CD8 microbeads (Miltenyi Biotec) were used to isolate CD8+T cells from PBMCs. Monocytes were seeded with GM-CSF (1000 U/ml) and IL-4 (800 U/ml) for 4 days to differentiate in monocyte-derived dendritic cells (moDCs). To stimulate peptide-specific T cells, monocyte-derived dendritic cells (moDCs) were pulsed with either peptides mix or with protein tumor lysates. When cultured with peptides mixes were composed as following: MIX A contained peptides from 1 to 5, while MIX B contained peptides from 6–10. When cultured with peptides, monocyte-derived dendritic cells (moDCs) were pulsed for 2 h at 37 °C with 10 µM peptide mix. Conversely, tumor lysates were obtained by resuspending each $1*10^6$ cells in 50 µl of PBS supplemented with protease inhibitors (A32955, Thermo Scientific Pierce, Waltham, MA). Cells were subsequently lysed by 5 cycles of freeze and thaw. Lysate was clarified by centrifuging it for 5 min at $500 \times g$ at 4 °C. Lysate was added on moDCs in a ratio of 1:10. Next, TNFa (10 ng/ml, Peprotech, Cranbury, NJ) and LPS (10 ng/ml) were added directly to moDC and incubated for 4 h to generate semi-mature moDC. Peptide-loaded semi-mature moDCs were then co-cultured with autologous purified CD8+ T cells at a 1:5 ratio in the presence of IL-21 (60 ng/ml, Peprotech). After 10–12 days, T cells were re-stimulated with autologous peptide-pulsed monocytes for an additional 10 days. Cultures were fed every 2–3 days as needed with either IL-2 (50 U/ml, STEMCELL Technologies) or IL-15 (10 ng/ml, R&D Systems).

## Human IFN-γ ELISpot assay

IFN-γ ELISpot assays were performed using commercially available human ELISpot reagent sets (ImmunoSpot, Bonn, Germany), accordingly to the manufacturer's instructions. When seeding PBMCs derived from healthy donors or patients, the maximum number of available cells (up to $6*10^5$) were seeded in each well. When working with expanded CD8+ T cells. $1.5*10^5$ T cells were seeded over CD14+ cells at a 10:1 (effector:target) ratio for each well. Seeded cells were stimulated in vitro with 20 ng/µl (2 ug/well) of each peptide at 37 °C for 72 h. After 3 days of stimulation, the number of cytokine-producing, antigen-specific T cells was evaluated using an ELISpot reader system (ImmunoSpot). SFU count recorded for each sample was subtracted of the background and, for PBMCs, data was normalized to SFU per $1*10^6$ seeded cells, whereas, for CD8+ T cells, data was normalized to SFU per $2.5*10^5$ seeded cells.

## Real time killing assay

To assess real time T cell killing of mesothelioma cell line (H28, H2452) we used the iCELLigence RTCA instrument (ACEA Biosciences). First,

50 µl of cell culturing media was added to each well of 16 well E-Plates (ACEA Biosciences) to measure the background impedance. Next, target cells were seeded at a density of 10,000 (H28) or 20,000 (H2452) cells/well of the E-Plate in a volume of 50 µl. The following day, when the Cell Index (CI) reached a level of around 1, effector cells (cultured CD8+ T cells) were added at an effector to target (E:T) ratio of 1:10 in 100 µl (reaching a total of 200 µl/well). Data recordings are shown as Normalized Cell Index.

## Mouse RNA-seq data

RNA-seq data from previously published study was retrieved from the EBI repository under the PRJEB15230 accession[45]. Retrieved sequencing data were analyzed using Chipster (https://chipster.csc.fi/) following the pipeline described in the publication.

## Surface plasmon resonance (SPR)

Poly K-tailed peptide interaction with the Adenoviral capsid was measured using SPR. Measurements were performed using the SPR Navi 420A instrument (Bionavis Ltd, Tampere, Finland). A constant flow rate of 20 µl/min was used throughout the experiments, and the temperature was set to +20 °C. Laser light with a wavelength of 670 nm was used for surface plasmon excitation and analysis. Gold sensors (BioNavis) were cleaned and activated by boiling for 5 min in a solution containing 5 ml MilliQ water, 1 ml hydroxide peroxide (Sigma), and 1 ml ammonium hydroxide (Sigma) with the gold-coated surface facing down. Next, sensors were removed from the cleaning solution and rinsed with abundant MilliQ water, and dried using a water aspirator and non-flammable dust remover. The glass surface of the activated sensors was cleaned using 70% EtOH solution before being placed in the instrument. Sensors were coated using 2 mg/ml linear-Polyethyleneimine (lPEI, Aldrich) dissolved in MilliQ water to make the gold surface positively charged. MilliQ water was used as a carrier in this step. Subsequently, the carrier solution was switched to PBS, and the fluidic system of the instrument and the sensor were washed. The VALO-mD901 viruses were immobilized in situ on the sensor surface by injecting -1.8 · $10^{11}$vp/ml in PBS (pH 7.4) for 10 min, followed by a 10 min wash with PBS. For testing the interaction between various peptides and the immobilized VALO-mD901 viruses, 100 µM of the tested peptides were injected onto the viruses.

## PeptiCRAd complex

PeptiCRAd complexes were prepared as previously described elsewhere[7,8,10,11]. Briefly, complexes were formed by mixing Ad5/3-D24 virus and peptides with a poly K tail (in PBS) at a ratio of 20 µg of peptides per $1 × 10^9$ VP per mouse in minimum volume. The mixture was incubated at room temperature for 15 min. Prior to injections, the complexes were further diluted with PBS to reach a final administration volume (50 µl per mouse). To combine two different poly K peptides in one complex, each PeptiCRAd was prepared individually: $5 × 10^8$ VP and 10 µg of each peptide were first complexed, then the two PeptiCRAd were combined after complexation, diluted, and injected.

## Animal experiments

Female Balb/cOlaHsd mice (4 to 6-week-old) were purchased from Envigo (Laboratory, Bar Harbor, Maine, UK). Mice were housed in individually ventilated cages (IVC) for a maximum of 5 mice per cage with food and water provided ad libitum and 12 h of light/dark cycle. Mice were monitored daily by animal caretakers. All injections and tumor measurements were performed under isoflurane anesthesia (Attane™ Vet). Mice were euthanized using $CO_2$, and death was confirmed by neck dislocation.

For the two pre-immunization experiments, mice ($n = 3$ per group) were allocated to five different groups and each mouse was administrated with three peptides in total, received as three separate injections in three different areas, one injection for each peptide. Each

injection contained 25 µg of individual peptide +25 µg of Poly(I:C) (HMW) VacciGrade (Invivogen; San Diego, CA), to a final injection volume of 50 µl. Additionally, two groups received 25 µg of Poly(I:C) in PBS (Adjuvant) or PBS only (Mock) as controls. For the first experiment, mice were immunized once a week for 2 weeks; after 7 days from the second vaccination, mice were euthanized, and spleens and lymph nodes were collected. In the second experiment, the prime and boosting were done respectively on days 0 and 21; on day 35, mice were euthanized, and spleens and lymph nodes were collected. In both experiments, the immunogenicity of each peptide was deconvoluted at a single mouse level by ELISpot assay.

For the vaccination with tumor lysate, mice ($n = 3$ per group, $n = 5$ for Lysate + Virus condition) were allocated into four different groups, and each mouse was immunized with a subcutaneous injection on the right flank. Tumor lysates were prepared following the protocol of Kawahara et al.[46]. Briefly, AB12 cells were detached, washed twice with PBS, and resuspended with 50 µl PBS for every $1*10^6$ cells. Next, the cell suspension was treated with 5 freeze-thaw cycles using liquid nitrogen and a water bath set at 37 °C. The lysates were stored at −80 °C until use. To prepare a combined adenovirus and tumor cell lysate vaccine, VALO-mD901virus was added to the lysate (after it was thoroughly vortexed) and stored on ice until administration. A total of $1 \times 10^6$ cell lysate plus $1 \times 10^9$ VP was subcutaneously injected in a final volume of 100 µl per mouse. As controls, AB12 cell lysate alone (Tumor Lysate group) or VALO-mD901virus alone (Virus Group) or PBS (Mock Group) was also prepared.

Mice were immunized on day 0 and received a boosting on day 21 (after 3 weeks). After 14 days from the second vaccination, mice were euthanized, and spleens and lymph nodes were collected. The immunogenicity of selected peptides was deconvoluted at a single mouse level by ELISpot assay.

For the assessment of efficacy of our selected immunogenic peptides to impact the engraftment and progression of AB12 murine mesothelioma cell line in vivo, we preimmunized a cohort of Balb/cOlaHsd mice with either PBS (Mock), Poly(I:C) (HMW) VacciGrade (Invivogen; San Diego, CA) (polyIC) or Poly(I:C) together with peptides 11 and 12 (poly IC + peptides). Mice were immunized using the long protocol reported before. Briefly, mice were primed once (day −35) and boosted 2 weeks after the last injection (day −14). Two weeks after the boost (day 0), mice were engrafted with $4 \times 10^6$ AB12 cells in 50 µl of un-supplemented RPMI, subcutaneously injected into the right flank. Tumor growth was followed using a digital caliper to measure vertical and horizontal dimensions of each tumor every 4–5 days. Tumor volume was calculated using the following formula:

$$Tumour\ volume = \frac{long\ measure \times (short\ \text{measure})^2}{2} \quad (2)$$

To assess the efficacy of our chosen immunogenic peptides complexed with VALO-mD901 virus (PeptiCRAd) in inducing an immune response capable of directly killing the AB12 murine mesothelioma cell line, we preimmunized a cohort of Balb/cOlaHsd mice. The mice were divided into three groups: one group received PBS (Mock), another group received VALO-mD901 virus alone (Virus Alone), and the third group received VALO-mD901 virus coated with peptides 11 and 12 (PeptiCRAd). Mice were immunized using subcutaneous injections. They were primed for three consecutive days (day 0, 1, and 2), followed by a booster injection 2 weeks later (day 16). Two weeks after the boost (day 23), mice were euthanized, and spleens were collected to perform a cell killing assay based on luciferase. Additional information related to the method is described below.

For in vivo assessment of PeptiCRAd´s ability to impact tumor growth, $4 \times 10^6$ AB12 cells in 50 µl of un-supplemented RPMI were injected subcutaneously into the right flank of 30 Balb/cOlaHsd mice. On day 19, post-tumor engraftment, mice were randomized and divided into three groups ($n = 10$ mice/group). Starting on day 21, mice were intratumorally treated three times every second day. PeptiCRAd was prepared as described above using Ad5/3-D24. Control groups received Ad5/3-D24 virus only in PBS (Virus Group) or PBS only (Mock Group). Tumor growth was followed as described above. Animals were sacrificed on day 45 post tumor injection. Spleens, tumors, and tumor-draining lymph nodes were collected for immunological analysis.

### Murine IFN-γ ELISpot assay
IFN-γ ELISpot assays were performed using commercially available mouse ELISpot reagent sets (ImmunoSpot, Bonn, Germany), accordingly to the manufacturer's instructions. Spleens, collected at the endpoint of the experiment, were reduced to single cell solution by passing them through a 70 µm strainer with the use of the back of a syringe plunger. Red blood cell cells were then lysed using ACK buffer (Gibco) following the manufacturer's instructions. Splenocytes were then resuspended in CTL test medium (ImmunoSpot, Bonn, Germany) and counted. For each well, $3 \times 10^5$ splenocytes were seeded and were stimulated with 20 ng/µl (2 µg/well in total) of each peptide at 37 °C for 72 h. After 3 days of stimulation, plates were developed, and spot count was obtained using CTL ImmunoSpot ELISpot plate reader system (ImmunoSpot, Bonn, Germany).

### Killing assay by LDH release
Killing assay was performed using LDH released by death cells using the CyQUANT™ LDH Cytotoxicity Assay kit (C20301, Thermo Fisher) using the manufacturer's instructions. Briefly, $15*10^4$ AB12 were seeded at day 0 in 100 µl. The following day, cohorts of immunized mice were euthanized, and spleens were collected and processed to reduce them to a single cell suspension. Splenocytes were then seeded at an effector to target (E:T) ratio of 50:1 in 100 µl (reaching a total of 200 µl/well). After 4 h or ON incubations, plates were centrifuged at $330 \times g$ for 5 min at RT and 50 µl of the supernatant were sampled for measuring LDH activity following the protocol included in the kit.

### Killing assay using Luciferin
AB12 cells line was transduced with a 3rd generation lentiviral vector carrying luciferase-eGFP genes separated by a P2A element previously produced[47] and kindly donated by Dr. Koski Jan and Dr. Korhonen Matti. A single pure clone was then derived using serial dilution form the pool of transduced cells. On day 0, $15*10^4$ AB12-Luc were seeded in 100 µl of each well of a white 96 well plate. The following day, cohorts of immunized mice were euthanized, and spleens were collected and processed to reduce them to a single cell suspension. Splenocytes were then seeded at an effector to target (E:T) ratio of 100:1 in 100 µl (reaching a total of 200 µl/well). After 4 h or ON incubations, plates were washed twice with PBS and then 150 µg/ml working solution of D-Luciferin (IVISbrite, 122799, PerkinElmer) in pre-warmed tissue culture medium were added to the remaining live cells, just prior sample acquisition. Luminescence data were acquired using Varioskan LUX (Thermo Fisher), 1 s long acquisition.

### Flow cytometry analysis
In the case of adherent cell culture, cells were detached either by scraping or by incubating them with PBS + 10 mM EDTA. Cells were stained using the following procedure: detached cells were centrifuged at $600 \times g$ for 5 min and washed twice with PBS. Cells were then incubated with either 1 µg per sample of TruStain FcX™ (anti-mouse CD16/32) Antibody (BioLegend, cat: 101320) or 5 µl per sample of Human TruStain FcX™ (Fc Receptor Blocking Solution) (BioLegend, cat: 422302) in cold for 10–15 min in accordance to the respective manufacturer instructions. Next, cells were stained with fluorochrome-labeled antibodies and incubated on ice for 30 min protected from light. Stained cells were then washed twice with PBS before sample acquisition.

Antibodies used in this study: APC anti-mouse CD3 (clone: 17A2, cat: 100236, Biolegend, 0.5 μg/1 million cells), FITC anti-mouse CD8a (clone: 53-6.7, cat: 100706, Biolegend, 1 μg/1 million cells), PerCP/cy5.5 anti-mouse CD107a (LAMP-1) (clone:1D4B, cat:121625, Biolegend, 5 μl/1 million cells), PE anti-mouse CD279 (PD-1) (clone:29 F.1A12, cat:135206, Biolegend, 1 μg/1 million cells), PerCP/cy5.5 anti-mouse CD366 (Tim-3) (clone:RMT3-23, cat:119718, Biolegend, 0.5 μg/1 million cells), PE-conjugated antihuman HLA-A, HLA-B, and HLA-C (clone W6/32, cat:311406, BioLegend, 5 μl/1 million cells), APC anti-mouse H2-Kd (clone:SF1-1.1, cat: 116619, Biolegend, 0.25 μg/1 million cells). All antibodies' mixes were incubated in a final volume of 100 μl and generally used in accordance to their respective manufacturer instructions.

Stained samples were acquired using a BD Accuri 6C Plus Flow Cytometer (BD Biosciences) and flow cytometric data were analyzed using FlowJo software v.10 (BD Life Sciences).

### Statistical analysis

GraphPad Prism 8.0 software (GraphPad Software, USA) or Python were used to perform statistical analysis. All results are expressed as the mean ± SEM. Additional information on the statistical test used can be found in the corresponding figure legend.

### Reporting summary

Further information on research design is available in the Nature Portfolio Reporting Summary linked to this article.

## Data availability

All the mass spectrometry proteomics data have been deposited to the ProteomeXchange Consortium via the PRIDE partner repository under the identifier PXD038273. The publicly available murine mesothelioma gene expression data used in this study are available in the EBI repository database under accession code PRJEB15230. The remaining data are available within the Article, Supplementary Information or Source Data file. Source data are provided with this paper.

## Code availability

Codes and custom scripts are available on the git hub repository (https://github.com/JacopoChiaro/PyptidomicsQC).

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

## Acknowledgements

We thank all the coauthors for their participation to this study. This work has been supported by European Research Council (ERC), Horizon 2020 (H2020) framework (Agreement No. 681219) (V.C.), Magnus Ehrnrooth Foundation (project No. 4706235) (V.C.), Jane and Aatos Erkko Foundation (Project No. 4705796) (V.C.), Finnish Cancer Foundation (project No. 4706116) (V.C.), Helsinki Institute of Life Science (HiLIFE) (project No. 797011004) (V.C.), Digital Precision Cancer Medicine Flagship iCAN (V.C.), GeneCellNano flagship (V.C.), Doctoral Programme in Drug Research (DPDR), Helsinki University (J.C.), Familjen Erling-Perssons stiftelse (20200922) (J.L., R.M.B.), Barncancerfonden (PR2019-0071) (J.L., R.M.B.), Swedish Research Council (2019-04830) (J.L., R.M.B.), Cancerfonden (CAN 2020, 20 1269 PjF) (J.L., R.M.B.), European Research Council (ERC) Horizon 2020 (847912 RESCUER and 965397 CCE_DART) (J.L., R.M.B.).

## Author contributions

J.C. and G.A. contributed equally to this work, S.F. and M.Fe. contributed equally to this work. Specific contributions: Conceptualization: J.C., G.A., S.F., and V.C. Investigation: J.C., G.A., S.F., M.Fe., H.C., V.F., D.C., S.R., M.Fu., F.H., and R.M.B. Data curation: J.C. and M.G. Formal analysis: J.C., G.A., V.F., and D.C. Software: J.C. Visualization: J.C. Project administration: J.C., S.F., and V.C. Writing—original draft: J.C., G.A., S.F., and M.G. Writing—review and editing: J.C., G.A., S.F., M.Fe., B.M., H.C., V.F., D.C., I.I., J.R., M.M., J.P., S.K., J.Ho., J.Ha., L.K., S.R., M.Fu., F.H., M.R., M.G., R.M.B., J.L., and V.C. Resources: I.I., J.R., M.M., J.P., S.K., J.Ho., J.Ha., L.K., and M.G. Funding acquisition: V.C.

## Competing interests

V.C. is co-founder and shareholder of Valo Therapeutics LTD. H.C. and L.K. are stakeholders of Valo Therapeutics LTD. All other named authors declare that they have no competing interests, financial or otherwise.

## Additional information

[1]Drug Research Program (DRP), ImmunoViroTherapy Lab (IVT), Division of Pharmaceutical Biosciences, Faculty of Pharmacy, University of Helsinki, Viikinkaari 5E, 00790 Helsinki, Finland. [2]Helsinki Institute of Life Science (HiLIFE), University of Helsinki, Fabianinkatu 33, 00710 Helsinki, Finland. [3]Translational Immunology Program (TRIMM), Faculty of Medicine Helsinki University, University of Helsinki, Haartmaninkatu 8, 00290 Helsinki, Finland. [4]Digital Precision Cancer Medicine Flagship (iCAN), University of Helsinki, 00014 Helsinki, Finland. [5]Institute for Molecular Medicine Finland (FIMM), HiLIFE, University of Helsinki, Helsinki, Finland. [6]Valo Therapeutics Oy, Viikinkaari 6, Helsinki, Finland, 00790 Helsinki, Finland. [7]Department of Biomedical Sciences, Humanitas University, Via Rita Levi Montalcini 4, 20090 Pieve Emanuele, MI, Italy. [8]Department of General Thoracic and Esophageal Surgery, Heart and Lung Center, Helsinki University Hospital, 00029 Helsinki, Finland. [9]Department of Surgery, Clinicum, University of Helsinki, 00029 Helsinki, Finland. [10]Department of Pathology, Helsinki University Hospital, Helsinki, Finland. [11]Research & Development Finnish Red Cross Blood Service Helsinki, Kivihaantie 7, 00310 Helsinki, Finland. [12]Finnish Red Cross Blood Service Biobank, Härkälenkki 13, 01730 Vantaa, Finland. [13]Department of Virology, National Institute of Public Health NIH—National Research Institute, 24 Chocimska Str., 00-791 Warsaw, Poland. [14]IRCCS Humanitas Research Hospital, Via Manzoni 56, 20089 Rozzano, MI, Italy. [15]Science for Life Laboratory, Department of Oncology-Pathology, Karolinska Institutet, Solna, Sweden. [16]Department of Molecular Medicine and Medical Biotechnology and CEINGE, Naples University Federico II, 80131 Naples, Italy. [17]These authors contributed equally: Jacopo Chiaro, Gabriella Antignani. ✉ e-mail: vincenzo.cerullo@helsinki.fi

