## [Peer Review File · Nature Communications]

Development of mesothelioma-specific oncolytic immunotherapy enabled by immunopeptidomimetics of murine and human mesothelioma tumorsREVIEWER COMMENTS

Reviewer #1 (Remarks to the Author): with expertise in mesothelioma, oncolytic therapy

Jacopo Chiaro et al studied the HLA class I immunopeptidomic of mesothelioma to use these peptides in combination with oncolytic adenovirus. They first identified class I HLA peptides from four human mesothelioma cell lines and two patients' biopsies by MHC immune-affinity purification, peptides elution, mass spectrometry and bioinformatics analysis. From the identified peptides pooled, they selected ten of them that were never studied in mesothelioma before to prime some healthy donor T cells using peptide pulsed autologous mature monocytes dendritic cells. They succeeded to amplify CD8+ T cells that recognize these peptides. They then characterize the HLA class I peptides from the murine mesothelioma cell lines AB12 by the same immunopeptidomic approach. They selected fifteen peptides, immunized mice against them and showed an expansion of murine T cells among splenocytes that recognized some of these peptides. Finally, they selected the two best peptides and added a poly-K tails to coat oncolytic adenovirus, an approach named peptiCRAD. They show that while the uncoated adenovirus is unable to reduce tumor development in vivo, peptiCRAD is able to block it. It is accompanied by a modification of T cells in the tumor microenvironment that display an activated phenotype.

The manuscript is well written and the results are convincing. The platform they described to identify tumor peptides and use them in combination with oncolytic adenovirus is innovative and original. However there are several minor issues to address.

Minor issues:

- Several figures (even the original figures) are not readable due to the use of too small caps such as Figure 3 and figure 5. Please increase caps size.
- On figure 2B, they identified longer peptides (12-16mers) from patients' biopsies than from mesothelioma cell lines. Do the authors have hypothesis to explain that? Could it be due to HLA class II peptides contamination that are known to be longer than class I peptides and that may come from HLA class II positive immune cells present in the tumor biopsy?
- On the figure 6A, to ease the reading, is it possible to align the "found" motives with the corresponding "expected" motives? Contaminant motives for Meso002 should not be

aligned with “expected” motives and put under and between C0701 and C0702.

- On figure 6B and 10E, please explain the legend “nB”, “WB” and “SB”.

- In figure 9, the authors selected 10 human mesothelioma peptides that appear to all have HLA-A*0201 binding motives, but they did not mention it. Was it part of the selection process? They then primed healthy donor T cells using peptides pulsed autologous moDCs without precising if the donor was selected for HLA-A*0201 expression. I guess it must have been the case. Can the authors precise that points.

Jean-Francois Fonteneau PhD,
INSERM, Nantes

Reviewer #2 (Remarks to the Author): with expertise in immunopeptidomics

Although the immunopeptidome data itself is useful and novel, as there are few (if any) immunopeptidome studies in this cancer type, this paper reads mostly about the immunopeptidome data QC and less so about the biology. The quality of the immunopeptidome data also appears to be below what one would expect in terms of replicate overlap. In terms of manuscript structure, this manuscript was incredible difficult to read and requires a major revision for clarity. At current, this manuscript does not have a clear big picture goal, and instead presents several smaller stories about HLA immunopeptidome data QC, a set of tools for immunopeptidome QC, and some immunological validation of potential T cell targets. I am not clear what the big picture question is? Is it how to QC immunopeptidome data? Is it trying to ID new antigens in mesothelioma-1 specific? The title and the paper itself do not match well, as I expected most of the paper to focus on the development of mesothelioma-1 specific oncolytic vaccine. I think this manuscript needs a major revision and several technical issues need to be addressed before it could be considered for publication.

Specific Comments:

- The writing of this manuscript is difficult to read. Many of the paragraphs are missing clear thesis sentences. Restructuring the introduction to present the challenge and how you

solved it would be helpful. At current, the introduction reads more like a stream of consciousness of thoughts.

- 10 figures are overwhelming and seems far too many for a Nature Communications article. Most of the figures are data QC, which I think can be condensed to the most useful panels into 1-2 figures max.

- Is this a high mutation tumor? Do you expect to see any neoantigens? I was surprised this was not discussed in the manuscript.

- PyptidOmicsQC seems more like a concatenation of scripts than a software tool. Is there a new QC that this group shows that is helpful for understanding the reproducibility of the data not used previously? If so, it would be helpful to explain this in the text.

- Why were these peptides analyzed for predicted binding using MHCflurry and not NetMHCpan? Is there an advantage of using one more the other. Adding this to the text would be helpful.

- Figure 3—overlap between the replicates is much lower than previous studies (most have 50-65% replicate overlap). Where these sample collected under different conditions? It would be helpful to show your replicate overlaps compared to those from previously published datasets.

- This paper feels more like execution of scripts for data QC and the authors did not see that there were issues in their data. It would be helpful to compare their results to published datasets to set thresholds for correlation and overlap of replicates for immunopeptidome samples.

- Figure 4—the correlation looks much lower than previous studies. Was this due to intended variation of the experiment or something else? Please see Figure 3, E & F in PMID: 29242379 for replicate correlations that appear much higher than those shown here.

- I am not sure how helpful Figure 5 with the clustering is. I can see value in this type of analyses to identify sample swaps, but I am less clear on its values of showing sample reproducibility.

- I may have missed it, but it was not clear how much w6/32 antibody and beads were used for the HLA immunopurification. What is the ratio? How was used for each sample? These details are needed to ensure reproducibility.

- It is typical for the reviewers to have access to the PXD022194 dataset using a password. This should be provided in the future. Along with the .d data from the timsTOF, the

database used for the search, and a sample file mapping list should be provided that specifically identifies each .d file with a sample annotation, the amount of sample used, the HLA typing, and other useful metadata. Please include all these items for reviewers to see prior to consideration for publication.

Reviewer #3 (Remarks to the Author): with expertise in immunopeptidomics

Summary of the study

Chiaro et al. proposed to identify new MHC-I cancer peptides from MPM tumors and to combine them with oncolytic virus-based vaccine therapy to boost an anti-tumoral T-cell response in vivo. For the identification of new tumor-associated antigens (TAA), they've characterized MPM-derived immunopeptidomes from mouse and human cell lines and 2 MPM human tumors. The authors have used conventional immunopurification to isolate MHC-class I peptides as well as standardized softwares (NetMHCpan, Gibbs cluster) to assess the quality of MHC-I eluted peptides. They've compared their data with MHC ligands eluted from healthy tissues from IEDB database and with gene expression transcriptomic databases to support and guide their findings. The authors have selected 2 peptides to test their potential anti-tumoral effect in a mouse model using the PeptiCRAD strategy consisting of an oncolytic adenovirus whose capsid is decorated with 2 poly-K MHC-I peptide candidates. The authors have observed a decreased tumor growth following treatment with PeptiCRAD as compared with the treatment with the virus alone. The authors have concluded that they've offered a proof of concept in mice that immunogenic peptides identified by immunopeptidomics can be used to generate cancer vaccines.

Major comment

The idea of boosting the immune system with oncovirus-derived vaccine appears to be a novel potential therapeutic route for treatment of tumors unresponsive to standard treatments. However, the major concern about the results presented in this manuscript is the lack of biological evidences supporting the immunogenic and tumor eradication dependent activities of the MHC-I peptides they've identified. If the rationale is based on

the fact that vaccination with cancer-specific peptides could boost tumor CD8+T cell responses, the results presented in the manuscript are not aligned with this idea. The data presented clearly showed that the virus alone is sufficient to induce the proliferation of T cells without any beneficial/additional effect observed when compared with mice treated with the PeptiCRAD (Figure 13). This raises important doubts about the functional and specific roles of the selected peptides to be immunogenic and to mount the anti-tumoral response observed in vivo (Figure 12c). If MHC-I peptide candidates are responsible for the T cell response, even partially, it has to be demonstrated. Key experiments meeting the expected standards of the field must be performed to prove the intrinsic immunogenic and more importantly, the anti-tumoral potential of the selected peptides alone. To further validate specificity of the immunogenic potential of peptides 11 and 12, experiments must include the quantification of these peptides in the immunopeptidomes isolated from the mouse cell line and the in vivo tumoral mouse model using heavy labeled peptides analyzed by MSMS based methods (PRM or SRM) (Bauer et al, Nat Comm 2022). Moreover, the quantification of HLA-tetramers/peptide 11-12 complexes recognized by T cells from PBMCs (HD or MESO patients) would add confidence in the peptide/HLA-I specificity predicted by their upstream informatic analysis. Most importantly, the measurement of anti-tumoral/cytotoxicity induced by a population of positive T-cells towards tumor cells displaying the HLA/peptide complex of interest would fortify the synergistic anti-tumoral effect claimed by the authors. Unfortunately, the lack of negative control peptides not predicted to generate any immunogenic/cytotoxic response (and not only the vehicle alone) in key experiments undermines the results presented. Similarly, testing PBMC from patients lacking the appropriate HLA would further support the entire study.

Minor comments

1. The first part of the manuscript describing the characterization of the immunopeptidomes from 4 human cell lines and 2 patient tumors is unnecessarily long and redundant with already reported data and represent mostly technical observations rather than new findings. Usually, the use of a positive immunopeptidome control (like JY cell line) along with eluted peptide length, predicted HLA binding affinity and peptide binding motif analyses are sufficient to rapidly assess the quality of a given immunopeptidome (Kovalchik et al. MCP

2021). In sum, the majority of the 8-12mers should correspond to weak or strong HLA predicted binders. Moreover, as mentioned by the authors, the immunopeptidomes of the 2 patient's tumors contain a lot of contaminants (longer peptides), further supporting the need to perform additional in vitro validation studies on particular peptide candidates. HLA binding affinity and peptide binding motifs are prediction-based analyses and in vitro studies should be rapidly undertaken to validate the immunogenicity of any MHC-I peptide candidates. The impact of over-represented HLA on the numbers of MHC-I peptides pooled down, the poor reproducibility between replicates following MSMS analysis by DDA are technical observations and they are not findings on their own. The link between the results exhibited in figures 3,4,5,7,8 with the scope of the manuscript is not obvious and we don't understand how it add to the final results. Also, the figure 6 is confusing and unreadable, especially 6B. If the Gibbs cluster cannot generate the expected consensus binding motif, it might simply reflect that the quality of the immunopeptidome is insufficient to generate significant results. A summary of the findings from figure 3 to 8 could have been represented using 1 figure only as in fig 10 for the mouse model.

2. Fig 9A lacks many essential details to make any conclusion: What is the CTL only? What is the positive control (positive peptide control) in the experiment? In the method section, it's mentioned that the peptide concentration used was 10mM, which is way too high. Usually, for this type of assay, we tend to use a range of 10nM to 100 uM (Yang et al. Nature Medecine, 2019). This could explain the similar response induced by all the peptides tested. Also, a negative peptide control is mandatory to demonstrate the specificity of the response. Flow cytometry results should also be shown with the minimum threshold (gating) established using a negative control as published elsewhere (Bear et al. Nature Communications 2021). For figure 9B, the number of replicates is not reported and the SEM are so large that nothing can be concluded from these results. Also, do PMBCs from HD and MESO exhibit HLA subtypes matching the predicted binding of peptides tested? If so, it should be labeled in the figure with the flow cytometry gating results. Based on the results shown, there are no significant differences between HD and MESO. Therefore, the conclusion (lines 362-364) is over-stated.

3. The in vivo model used for the application of vaccine-based therapy seems to represent a good model since the immunopeptidome of AB12 cell line display a greater quality of MHC-I peptides. A table showing all the criteria used to filter the list of peptide candidates should

be presented, like the log₂FC of gene expression, predicted binding affinity by at least 2 algorithms, the peptide sequences and the HLA subtype predicted to bind the peptides. Additionally, a schematic of the bioinformatic pipeline used to end up with the 10 MPM mouse peptides selected from gene expression databases and cell lines with all the filtering criteria applied would add clarity for the reader.

4. MHC class I expression at the cell surface of the MESO tumors from the patients should be quantified as performed with the other cell line. The results from this quantification could help to explain the poor quality of their corresponding immunopeptidomes.

5. It would be important to report mouse toxicity measurement (for instance weight measurement) for the in vivo model when monitoring the tumor growth under different experimental conditions.

6. A clearer hypothesis and rationale should be described and clarified in the introduction of the manuscript. In the abstract, the authors mentioned that MPM is responsive to immunotherapeutic cancers (line 44). However, in the introduction, the authors commented that ICI monotherapy has limited impact in MPM patients and “MPM are not ‘hot’ tumors, i.e. infiltrated with a of lot of T cells, but still infiltrated with T cells”. Moreover, the authors cited study reporting the low mutational burden of MPM. Therefore, the rationale of identifying new antigens and more importantly the origin of MHC-I peptides (self or non-self, mutated or over-expressed) should be mentioned in the beginning of the manuscript. The overview of the current knowledge of antigens associated to MPM should be described with the rationale. This type of cited studies would guide the reader to understand the selection of peptides based on transcriptomic and not exome sequencing data for instance.

References

Bauer, J., Köhler, N., Maringer, Y. et al. The oncogenic fusion protein DNAJB1-PRKACA can be specifically targeted by peptide-based immunotherapy in fibrolamellar hepatocellular carcinoma. *Nat Commun* 13, 6401 (2022). <https://doi.org/10.1038/s41467-022-33746-3>

Kevin A. Kovalchik et al. MhcVizPipe: A Quality Control Software for Rapid Assessment of Small- to Large-Scale Immunopeptidome Datasets, *Molecular & Cellular Proteomics*, Volume

21, Issue 1, 2022, 100178, ISSN 1535-9476, <https://doi.org/10.1016/j.mcpro.2021.100178>.

Yang W et al. Immunogenic neoantigens derived from gene fusions stimulate T cell responses. *Nat Med*. 2019 May;25(5):767-775. doi: 10.1038/s41591-019-0434-2. Epub 2019 Apr 22. PMID: 31011208; PMCID: PMC6558662.

Bear, A.S., Blanchard, T., Cesare, J. et al. Biochemical and functional characterization of mutant KRAS epitopes validates this oncoprotein for immunological targeting. *Nat Commun* 12, 4365 (2021). <https://doi.org/10.1038/s41467-021-24562-2>

Manuscript revised by:

Isabelle Sirois PhD

Senior scientist at CaronLab

Reviewers' Comments:

Reviewer #1 (Remarks to the Author): **Jean-Francois Fonteneau**
with expertise in mesothelioma, oncolytic therapy

Jacopo Chiaro et al studied the HLA class I immunopeptidomic of mesothelioma to use these peptides in combination with oncolytic adenovirus. They first identified class I HLA peptides from four human mesothelioma cell lines and two patients' biopsies by MHC immune-affinity purification, peptides elution, mass spectrometry and bioinformatics analysis. From the identified peptides pooled, they selected ten of them that were never studied in mesothelioma before to prime some healthy donor T cells using peptide pulsed autologous mature monocytes dendritic cells. They succeeded to amplify CD8+ T cells that recognize these peptides. They then characterize the HLA class I peptides from the murine mesothelioma cell lines AB12 by the same immunopeptidomic approach. They selected fifteen peptides, immunized mice against them and showed an expansion of murine T cells among splenocytes that recognized some of these peptides. Finally, they selected the two best peptides and added a poly-K tails to coat oncolytic adenovirus, an approach named peptiCRAD. They show that while the uncoated adenovirus is unable to reduce tumor development in vivo, peptiCRAD is able to block it. It is accompanied by a modification of T cells in the tumor microenvironment that display an activated phenotype.

The manuscript is well written and the results are convincing. The platform they described to identify tumor peptides and use them in combination with oncolytic adenovirus is innovative and original. However there are several minor issues to address.

We thank the review very much for the appreciation of our work and for the suggestions to improve it. We have taken all the suggestions into account and consequently changed the manuscript. Below it can be found the point-to-point responses to the reviewer's concern. We have highlighted the sections where the manuscript was modified.

Minor issues:

1) Several figures (even the original figures) are not readable due to the use of too small caps such as Figure 3 and figure 5. Please increase caps size.

We agree the figures were indeed difficult to read. To follow up to the concerns of other reviewers and in an attempt of reducing the number of figures, the one herein referred as **Figure 3** by the reviewer, has been removed from the updated version of the manuscript. On the other hand, what is herein referred as **Figure 5** has been updated and its readability has been improved. The new version of the figure is available in the updated version of the manuscript as new **Figure 3** and below.

Figure 1: Updated figure about sample clustering based on immunopeptidomics-derived peptides and HLA haplotype composition.

2) On figure 2B, they identified longer peptides (12-16mers) from patients' biopsies than from mesothelioma cell lines. Do the authors have hypothesis to explain that? Could it be due to HLA class II peptides contamination that are known to be longer than class I peptides and that may come from HLA class II positive immune cells present in the tumor biopsy?

This is indeed a very good point, following up on this comment we investigated more in depth this enrichment in longer peptides observed in some of the immunopeptidomics run. We tend to exclude the significant presence of MHC class II peptides. We have observed a peptide enrichment similar to what has been observed by Fritsche et al. (1). In their work, the authors have attributed the presence of longer peptides to the effect of proteolytic activity, which they have shown to look like peptides ladders as we also observe in our samples (Figure 2A). Interestingly, by computing what they refer to as "protein score" and applying a threshold corresponding to the 95th percentile of the distributions obtained by good quality runs (Figure 2B) we were able to significantly improve the quality our immunopeptidomics run (Figure 2C) and eliminate the majority of the contaminant peptides (Figure 2D).

Figure 2: MESO002 immunopeptidomic dataset cleanup. A) example of a “peptide ladder” suggesting that longer peptides are the effect of proteolytic activity. B) Distribution of the “protein coverage” scores computed for each source protein of the eluted peptides found in each immunopeptidomics run. The dotted line represents the average 95th percentile of the runs corresponding to the mesothelioma cell lines (which we considered of high quality). This threshold was used to determine “high quality” and “low quality” peptides.

These data are currently not present in the updated version of the manuscript, as they, despite being very interesting, could divert the article from its original focus (a concern expressed by other reviewers). However, they can be integrated in the article upon request of either the reviewer or the editor.

3) On the figure 6A, to ease the reading, is it possible to align the “found” motives with the corresponding “expected” motives? Contaminant motives for Meso002 should not be aligned with “expected” motives and put under and between C0701 and C0702.

This is indeed a good observation. A new version of the figure generated by following the reviewer’s suggestion is now available in the updated version of manuscript, there referred as **Figure 2**. Notably, after the reanalysis (cleanup) of the MESO002 patient immunopeptidomic dataset, the contaminant motif has now disappeared. Please see below.

Figure 3: Updated figure about eluted peptides binding specificity deconvolution

4) On figure 6B and 10E, please explain the legend “nB”, “WB” and “SB”.

We thank the reviewer for the comment as this has indeed been a lack on our side. An updated explanation for “nB”, “WB” and “SB” is now available in the corresponding figure legend. Please see the detail of the updated Figure legend below:

“Stacked bar plot illustrating the outcome of peptide-specificity deconvolution based on machine learning-based prediction performed using MHCflurry. Peptide specificity is annotated by using the best predicted peptide:MHC allele per each peptide. In the plot, the abbreviation “nB” represents “non-Binders,” “WB” stands for “Weak Binders,” and “SB” signifies “Strong Binders.””

5) In figure 9, the authors selected 10 human mesothelioma peptides that appear to all have HLA-A*0201 binding motives, but they did not mention it. Was it part of the selection process? They then primed healthy donor T cells using peptides pulsed autologous moDCs without precising if the donor was selected for HLA-A*0201 expression. I guess it must have been the case. Can the authors precise that points.

We thank the reviewer for having highlighted an unclear section in the manuscript. We have rewritten that part and updated the manuscript at lines: 258-265

Briefly, all human peptides but the number 7,8 and 10 have HLA-A2 specificity (either HLA-A02:01, A02:02 or A02:05 which all have similar/identical binding motifs).

Peptides 7 and 8, on the other hand, have binding specificity for the HLA-B08:01, while peptide 10 for HLA-C07:02 or C06:01.

All the healthy donors’ PBMCs used in the study had previously been HLA typed to ensure that all the peptides could be effectively used for each subject.

Supplementary table S1 has been produced to clarify this section and has been added to the manuscript (line 263).

Jean-Francois Fonteneau PhD,

#####

Reviewer #2 (Remarks to the Author): with expertise in immunopeptidomics

Although the immunopeptidome data itself is useful and novel, as there are few (if any) immunopeptidome studies in this cancer type, this paper reads mostly about the immunopeptidome data QC and less so about the biology. The quality of the immunopeptidome data also appears to be below what one would expect in terms of replicate overlap.

In terms of manuscript structure, this manuscript was incredible difficult to read and requires a major revision for clarity. At current, this manuscript does not have a clear big picture goal, and instead presents several smaller stories about HLA immunopeptidome data QC, a set of tools for immunopeptidome QC, and some immunological validation of potential T cell targets.

I am not clear what the big picture question is? Is it how to QC immunopeptidome data? Is it trying to ID new antigens in mesothelioma-1 specific? The title and the paper itself do not match well, as I expected most of the paper to focus on the development of mesothelioma-1 specific oncolytic vaccine. I think this manuscript needs a major revision and several technical issues need to be addressed before it could be considered for publication.

We agree and thank the reviewer for this comment. We have significantly improved the manuscript and its experimental part to highlight the real purpose of it, i.e. the use of the immunopeptidomics to develop and formulate precision cancer oncolytic immunotherapy.

Specific Comments:

1) The writing of this manuscript is difficult to read. Many of the paragraphs are missing clear thesis sentences. Restructuring the introduction to present the challenge and how you solved it would be helpful. At current, the introduction reads more like a stream of consciousness of thoughts.

We agree with the reviewer that the introduction required a major revision to improve its clarity. Therefore, the introduction has been completely revised and updated in the new version of the manuscript.

2) 10 figures are overwhelming and seems far too many for a Nature Communications article. Most of the figures are data QC, which I think can be condensed to the most useful panels into 1-2 figures max.

We agree with the reviewer, indeed, the final number of figures now account for 6 in total. All the data relative to the QC was condensed in 2 Figures (Figure 1 and 2) as it was done for the section regarding the work in mice. With the aim of reducing the number of figures, the *in vivo* validation of the selected peptides was overall condensed into 2 Figures as well (Figure 5 and 6).

3) Is this a high mutation tumor? Do you expect to see any neoantigens? I was surprised this was not discussed in the manuscript.

In the discussion section of the manuscript, we stated that mesothelioma has low mutational burden (lines 477-484), and, for this reason, it was more relevant to find immunotherapeutic strategies that were focusing on tumor associated antigens (TAA). However, we realize that the message may not have been effectively conveyed in the previous version of the manuscript. To address this concern and further strengthen the rationale of our work, we have added an additional statement in the introduction (lines 110 - 112) specifically addressing the relevance of targeting TAA in mesothelioma. We believe that this revision will better emphasize the significance of our research in the context of mesothelioma's low mutational burden.

4) PyptidOmicsQC seems more like a concatenation of scripts than a software tool. Is there a new QC that this group shows that is helpful for understanding the reproducibility of the data not used previously? If so, it would be helpful to explain this in the text.

This is indeed a good point. At the moment, PyptidOmicsQC is a python script running locally. Aside of speeding up the quality control of immunopeptidomics raw data files, the only novelty is represented by the visualization we propose in the updated **Figure 2**, where our stacked bar graphs represent, not only an effective way to visualize peptide deconvolution but also gives an immediate glimpse of the most favorite HLAs used by a specific tumor/cell line and, ultimately, give an idea of the level of contaminant peptides in each sample.

The aim would be for the future to develop an easily accessible web application for PyptidOmicsQC. To respond to the reviewer's comment, we have modified in the manuscript the way we address to the tool, and we now refer to it as script.

5) Why were these peptides analyzed for predicted binding using MHCflurry and not NetMHCpan? Is there an advantage of using one more the other. Adding this to the text would be helpful.

Peptide MHC-binding affinity prediction was performed using MHCflurry rather than NetMHCpan, not only because it showed to have a better performance (2), but also because MHCflurry is an open-source software coded in Python and available as a PyPI package. This makes it much more versatile and easier to integrate in other Python scripts/work environments such as, for instance, PyptidOmicsQC. We thank the reviewer for this comment as we overlooked the issue, and we have not properly addressed it in the previous version of the manuscript. We now have clarified the issue in the Methods section of the updated version of the manuscript (lines 623-626).

6) Figure 3—overlap between the replicates is much lower than previous studies (most have 50-65% replicate overlap). Where these sample collected under different conditions? It would be helpful to show your replicate overlaps compared to those from previously published datasets.

We agree with the reviewer that for some of the biological replicates we obtained rather low peptide overlap, although we identified the problem and it appeared to be mainly technical. For the samples presenting a lower quality, new replicates were performed. However, as we have substantially increased the number of replicates, a figure showing the overlap of all our datasets appeared very confusing. Additionally, in order to reduce the number of figures regarding the QC, as requested by the reviewer, we thought of removing that Figure from the manuscript.

Lastly, peptide selection for further immunological validation was performed, taking into account both the frequency with which a certain peptide had been identified and the peptide:MHC binding affinity. Only frequently observed peptides and "strong binders" were considered as putative candidate peptides for further immunological validation.

7) This paper feels more like execution of scripts for data QC and the authors did not see that there were issues in their data. It would be helpful to compare their results to published datasets to set thresholds for correlation and overlap of replicates for immunopeptidome samples.

We acknowledge the reviewer's comment regarding the excessive emphasis on quality control (QC) in the previous version of the manuscript. We have made revisions to address this concern and have taken steps to improve the quality of our data. We increased the number of replicates and made improvements in the sample preparation for mass spectrometry as well as on the mass spectrometry side itself.

Nonetheless, the **updated Figure 1** demonstrates that the distribution of peptide lengths among the eluted peptides aligns with the expectations. Additionally, when utilizing machine learning peptide:MHC binding affinity prediction, we found that, on average, over 90% of the 9mers from each sample were considered “binders” for the expected MHC alleles of the corresponding samples. Furthermore, as shown in the **updated Figure 2**, the obtained binding motifs corresponded to the expected patterns. These results indicate that our experimental runs were not of poor quality.

We believe that another factor contributing to the limited overlap might be the sampling depth. The peptide repertoire presented by the cells used in our study might potentially be extensive, and sampling only thousands of peptides at a time may represent a small fraction of the overall peptidome landscape. Consequently, a greater number of replicates would be required to achieve a more comprehensive coverage of the entire peptidome landscape.

8) Figure 4—the correlation looks much lower than previous studies. Was this due to intended variation of the experiment or something else? Please see Figure 3, E & F in PMID: 29242379 for replicate correlations that appear much higher than those shown here.

We acknowledge the reviewer's observation regarding the significant differences in correlation between replicates in our samples compared to those presented by Chong et al.

However, we would like to point out that there are several potential factors that could contribute to the low spectral intensity correlation among replicates in DDA immunopeptidomics runs:

1. **Biological variability:** The presentation of peptides by the MHC can vary between different biological replicates due to inherent biological variability, such as differences in protein expression, cellular processes, or post-translational modifications. This variability can result in low correlation of spectral intensities between replicates.
2. **Technical variability:** Immunopeptidomics experiments involve multiple experimental steps, including sample preparation, peptide extraction, purification, and mass spectrometry analysis. Variability in any of these steps can contribute to differences in spectral intensities between replicates. Factors such as sample handling, instrument performance, or experimental conditions can introduce technical variability, leading to lower correlation between replicates.
3. **Noise and measurement limitations:** Mass spectrometry-based techniques have inherent noise and limitations. Variability in ionization efficiency, fragmentation, or detection can introduce noise into the spectral intensity measurements, reducing the correlation between replicates.
4. **Data preprocessing and normalization:** The choice of data preprocessing and normalization methods can also affect the correlation between spectral intensities.

In particular, we believe that any variations in these steps may have contributed to the difference observed between our data and the ones presented by Chong et al., who undoubtedly reported exceptionally robust runs.

Nevertheless, we are highly confident in the quality of our runs, as indicated by other parameters that meet the standards set by the literature, including:

1. The distribution of peptide lengths aligns with expectations.
2. A high number of MHC-specific peptides are detected, either within the 8-13mers range or specifically focusing on 9mers.
3. The identified MHC-binding motifs of eluted peptides correspond to the anticipated patterns of the alleles present in the respective samples.

9) I am not sure how helpful Figure 5 with the clustering is. I can see value in this type of analyses to identify sample swaps, but I am less clear on its values of showing sample reproducibility.

We apologize for the lack of clarity in this particular section of the manuscript. We would like to clarify that the intention of **Figure 5** was not to address sample reproducibility, but rather to highlight the significant role of MHC haplotype composition in shaping the repertoire of presented epitopes. Our objective was to demonstrate that the overlap between immunopeptidomes of different samples increases in proportion to the number of shared MHCs or binding motifs.

This phenomenon was observed in a larger scale by Marcu and colleagues as well. In their study, they performed immunopeptidomics on several healthy tissues from different subjects, and they reported a higher similarity between the immunopeptidomes (as well as the composition of source proteins) in samples originating from the same subjects compared to samples from the same tissue types but from different subjects (3).

However, we acknowledge that panel A in the previous figure was redundant and did not effectively convey the intended message. As a result, it has been omitted from the updated Figure 3 in the revised version of the manuscript. If the reviewer requests it, this figure could be relocated to the supplementary material.

We have made efforts to address and clarify this section in the updated version of the manuscript. Please refer to lines 189-200.

Previous Version of the Figure. Panel A was considered superfluous and has been removed to improve readability of the updated figure (see below) and clarity in the text.

Updated figure present in the new revised version of the manuscript.

10) I may have missed it, but it was not clear how much w6/32 antibody and beads were used for the HLA immunopurification. What is the ratio? How was used for each sample? These details are needed to ensure reproducibility.

We thank the reviewer for the comment, as we did not realize this information was missing from the manuscript. We used 1mg of W6/32 antibody (*inVivo*MAb, BioXCell) with 2mL of pre-packed Agarose Resin present in the AminoLink™ column (AminoLink Plus Immobilization kit, Thermo-Fischer). Antibody linking was performed according to manufacturer instruction using the linking procedure at neutral pH (pH=7.2). We updated the manuscript with the missing information at lines 541-543.

11) It is typical for the reviewers to have access to the PXD022194 dataset using a password. This should be provided in the future. Along with the .d data from the timsTOF, the database used for the search, and a sample file mapping list should be provided that specifically identifies each .d file with a sample annotation, the amount of sample used, the HLA typing, and other useful metadata. Please include all these items for reviewers to see prior to consideration for publication.

We apologize the reviewer for the inconvenience, unfortunately the reported PRIDE identifier (PXD022194) was linking to the wrong page/datasets. The correct PRIDE id is now reported in the updated version of the manuscript.

All the datasets are now uploaded in ProteomeXchange/PRIDE.

Project accession: PXD038273

Reviewers should visit the website at the following link:

<https://www.ebi.ac.uk/pride/login>

And enter with the following:

Username: reviewer_pxd038273@ebi.ac.uk

Password: Y0yz7Zpl

#####

Reviewer #3 (Remarks to the Author): **Isabelle Sirois PhD**

with expertise in immunopeptidomics

Summary of the study

Chiaro et al. proposed to identify new MHC-I cancer peptides from MPM tumors and to combine them with oncolytic virus-based vaccine therapy to boost an anti-tumoral T-cell response in vivo. For the identification of new tumor-associated antigens (TAA), they've characterized MPM-derived immunopeptidomes from mouse and human cell lines and 2 MPM human tumors. The authors have used conventional immunopurification to isolate MHC-class I peptides as well as standardized softwares (NetMHCpan, Gibbs cluster) to assess the quality of MHC-I eluted peptides. They've compared their data with MHC ligands eluted from healthy tissues from IEDB database and with gene expression transcriptomic databases to support and guide their findings. The authors have selected 2 peptides to test their potential anti-tumoral effect in a mouse model using the PeptiCRAD strategy consisting of an oncolytic adenovirus whose capsid is decorated with 2 poly-K MHC-I peptide candidates. The authors have observed a decreased tumor growth following treatment with PeptiCRAD as compared with the treatment with the virus alone. The authors have concluded that they've offered a proof of concept in mice that immunogenic peptides identified by immunopeptidomics can be used to generate cancer vaccines.

We thank the reviewer for the comment. The manuscript has been significantly changed to highlight better the underpinning message of utilizing the immunopeptidomic analysis as bases for precision cancer oncolytic immunotherapy.

Major comment

1) The idea of boosting the immune system with oncovirus-derived vaccine appears to be a novel potential therapeutic route for treatment of tumors unresponsive to standard treatments. However, the major concern about the results presented in this manuscript is the lack of biological evidences supporting the immunogenic and tumor eradication dependent activities of the MHC-I peptides they've identified. If the rationale is based on the fact that vaccination with cancer-specific peptides could boost tumor CD8+T cell responses, the results presented in the manuscript are not aligned with this idea. The data presented clearly showed that the virus alone is sufficient to induce the proliferation of T cells without any beneficial/additional effect observed when compared with mice treated with the PeptiCRAD (Figure 13).

This raises important doubts about the functional and specific roles of the selected peptides to be immunogenic and to mount the anti-tumoral response observed in vivo (Figure 12c).

If MHC-I peptide candidates are responsible for the T cell response, even partially, it has to be demonstrated. Key experiments meeting the expected standards of the field must be performed to prove the intrinsic immunogenic and more importantly, the anti-tumoral potential of the selected peptides alone.

To further validate specificity of the immunogenic potential of peptides 11 and 12, experiments must include the quantification of these peptides in the immunopeptidomes isolated from the mouse cell line and the in vivo tumoral mouse model using heavy labeled peptides analyzed by MSMS based methods (PRM or SRM) (Bauer et al, Nat Comm 2022).

We apologize for the lack of clarity in these sections of the manuscript. When discussing immunogenicity, we refer to the ability of a specific moiety or protein product to elicit an immune response. Therefore, we do not see how the quantification of peptides 11 and 12 can further validate their immunogenicity. The approach suggested by the reviewer is typically employed to confirm the presence of peptides derived from aberrant or non-canonical protein products. In our case, finding these selected peptides in four biological replicates, with high predicted MHC binding affinity and derived from abundant proteins in the source cell line, serves as strong evidence to proceed with their immunological validation.

Furthermore, we believe that we have met the standard for peptide validation in the literature as we have demonstrated the immunogenic potential of peptides 11 and 12 in multiple ways. Firstly, we observed their capacity to specifically activate T cells through vaccination using an immune-adjuvant with different protocols and time schedules (see Updated Figure 5E-H). Additionally, splenocytes derived from mice immunized with our peptides showed significant killing activity when co-incubated with AB12 mesothelioma tumor cells (see Updated Figure 5I). Moreover, we performed an immunization with a whole tumor lysate and observed a response towards our selected peptides of interest (see Updated Figure 5J, K). Finally, we immunized a group of mice with peptides 11 and 12 (as shown in Updated Figure 5L) and subsequently engrafted the mice with AB12 cells, where we observed improved tumor control in the mice immunized with our selected peptides (see Updated Figure 5M). These data sufficiently demonstrate the anti-tumor activity of our selected peptides.

Regarding the reviewer's concern about the efficacy of the intratumorally injected Adenovirus alone it is undeniable and somehow expected, given that we are intratumorally administering a pathogen in immune-competent animals.

However, the superior efficacy of PeptiCRAAd compared to the controls used in the experiment may have been misunderstood due to ineffective data visualization. As shown in the updated Figure 6E, the average tumor volumes of the "Virus alone" and "Mock" groups are very similar. We believe that using the average "Average Tumor Volume" of the "Virus alone" group as a threshold for defining the therapeutic success rate in the subsequent analysis has been misleading. By using the "Median Volume" as the threshold, the message is better conveyed, as the therapeutic success rate is now 44% for both the Mock and Virus alone groups, while it is 89% for the PeptiCRAAd group (see below and updated Figure 6F). These data align better with the findings shown in the previous panel, the average tumor growth curves.

Furthermore, these data support what we observed within the tumor microenvironment, where T cells in the PeptiCRAAd group exhibit increased killing and exhaustion markers (see Figure 6H, I). We believe that this profile is a consequence of a strong antitumor effect that unfortunately showed signs of exhaustion by day 45 from tumor engraftment.

To ultimately demonstrate the effect of PeptiCRAd in promoting the killing of AB12 target cancer cells, we opted for an ex vivo approach. We immunized three groups of mice with PBS (Mock group), VALOm901 (Virus alone) or PeptiCRAd, which consists of VALOm901 complexed with peptides 11 and 12 (PeptiCRAd group). We then used the splenocytes from these animals to perform a killing assay using AB12 cells that stably express luciferase (AB12 Luc). Cell viability at the designated endpoint was assessed using luciferin. Our data show that splenocytes from mice pre-immunized with Virus alone did not exhibit any specific killing capacity when co-cultured with AB12-Luc cells. Conversely, splenocytes from mice pre-immunized with PeptiCRAd demonstrated higher killing capacity (see **Updated Figure 6 B, C**).

2) Moreover, the quantification of HLA-tetramers/peptide 11-12 complexes recognized by T cells from PBMCs (HD or MESO patients) would add confidence in the peptide/HLA-I specificity predicted by their upstream informatic analysis.

Unfortunately, peptides 11 and 12 used in vivo could not be tested in human subjects as these peptides are binders of the murine MHC H2-Kd which cannot be found in human. For this reason, the section related to the in vivo work is more intended to be a proof of concept of the pipeline which goes from the discovery of the peptides to the formulation of a vaccine.

3) Most importantly, the measurement of anti-tumoral/cytotoxicity induced by a population of positive T-cells towards tumor cells displaying the HLA/peptide complex of interest would fortify the synergistic anti-tumoral effect claimed by the authors.

We thank the reviewer for the comment, and to address the reviewer's concern we sought to assess the functional activity of our selected peptides by performing killing assays using tumors included in the immunopeptidomics study as target cells.

In the murine context we have shown twice the killing capacity promoted by our selected peptides see **Updated Figure 5I, M** and **Updated Figure 6C**.

While in human setting we have shown that T cells cocultured with our selected mix of peptides showed superior killing capacity of 2 mesothelioma cell lines compared to T cells expanded using MART-1 as irrelevant peptide (see **updated Figure 4F**).

4) Unfortunately, the lack of negative control peptides not predicted to generate any immunogenic/cytotoxic response (and not only the vehicle alone) in key experiments undermines the results presented.

We acknowledge the reviewer's suggestion regarding the inclusion of a group of irrelevant peptides to our experiment would have strengthened our findings.

However, we believe that in the previous section of our work the focus had been more directed towards the immunogenic and cytotoxic characterization of our selected peptides. Therefore, in the experimental setting presented in the updated Figure 6, our primary focus was to assess the efficacy of our coated-Adenovirus platform in controlling tumor progression rather than solely evaluating the peptides themselves. Ideally, we would have included the following groups in our experiment: 1) PBS, 2) Virus Alone, 3) Peptides Alone, and 4) PeptiCRAd.

However, previous studies by Capasso et al. and Ylösmäki et al. (4, 5) (in their Figure 5A, Figure 4A respectively), have shown that intratumoral administration of peptides alone, without any additional immune adjuvant, only leads to minor effect in tumor control when compared to the "Mock" or "Vehicle Alone" groups. Considering these findings, we decided to bypass the "Peptides Alone" group and instead included the following groups: 1) Mock, 2) Vehicle (Uncoated Adenovirus), and 3) PeptiCRAd (Peptide-Coated Adenovirus).

5) Similarly, testing PBMC from patients lacking the appropriate HLA would further support the entire study.

We acknowledge the reviewer's suggestion for conducting the proposed experiment. However, it can be challenging to account for unspecific responses in such settings, as peptides may be sub-optimally presented and generate unexpected immune responses.

In our study, we conducted peptide validation using a cohort of 11 different subjects. Within this cohort, we observed variations in the immunogenicity of the peptides, with some peptides rarely inducing any interferon gamma secretion, while others were more frequently able to promote T cell activation. For example, peptide 2 (RLASYLDKV) demonstrated limited immunogenicity, whereas peptide 9 (RLASYLDRV), which shares the same HLA specificity but differs by only one amino acid in a non-anchor position, exhibited a distinct stronger immunogenicity profile (see updated Figure 4 B,C,D).

Minor comments

1) The first part of the manuscript describing the characterization of the immunopeptidomes from 4 human cell lines and 2 patient tumors is unnecessarily long and redundant with already reported data and represent mostly technical observations rather than new findings.

- 1.1) Usually, the use of a positive immunopeptidome control (like JY cell line) along with eluted peptide length, predicted HLA binding affinity and peptide binding motif analyses are sufficient to rapidly assess the quality of a given immunopeptidome (Kovalchik et al. MCP 2021). In sum, the majority of the 8-12mers should correspond to weak or strong HLA predicted binders.**
- 1.2) Moreover, as mentioned by the authors, the immunopeptidomes of the 2 patient's tumors contain a lot of contaminants (longer peptides), further supporting the need to perform additional in vitro validation studies on particular peptide candidates.**
- 1.3) HLA binding affinity and peptide binding motifs are prediction-based analyses and in vitro studies should be rapidly undertaken to validate the immunogenicity of any MHC-I peptide candidates.**

We appreciate the reviewer's suggestion regarding the use of a cell line, such as JY, as a reference to assess the quality of our study. However, to enhance the quality of our data, we conducted additional replicates for three cell lines (H28, 211H, H2452) with minor modifications to the sample preparation protocol.

When it was not feasible to increase the number of replicates due to lack of additional sample material (MESO002), we conducted an in-silico identification and elimination of contaminant peptides. This process, which is explained in more detail above, was carried out following the instructions provided in the work of Fritsche et al. (1).

Ultimately, following the reviewer's recommendation for good quality criteria, we observed that our eluted peptides exhibited the typical length distribution commonly observed in other immunopeptidomics studies. Furthermore, we employed a machine learning peptide:MHC binding affinity prediction tool and found that, on average, over 80% of the 8-13mers in our dataset were classified as specific "Binders" for the corresponding MHC alleles expected in the respective samples. Moreover, when considering only the 9mers from each sample, the number of identified "binders" exceeded 90%.

Additionally, as depicted in the updated Figure 2, the obtained binding motifs aligned with the expected patterns. These findings suggest that our experimental runs were not of poor quality and that our data align with the current literature.

We have incorporated these results into the revised version of the manuscript to further support the reliability and quality of our data.

1.4) The impact of over-represented HLA on the numbers of MHC-I peptides pooled down, the poor reproducibility between replicates following MSMS analysis by DDA are technical observations and they are not findings on their own.

1.5) The link between the results exhibited in figures 3,4,5,7,8 with the scope of the manuscript is not obvious and we don't understand how it add to the final results. A summary of the findings from figure 3 to 8 could have been represented using 1 figure only as in fig 10 for the mouse model.

We agree with the reviewer that the findings mentioned are not in line with the scope of identification of epitopes for the development of a therapeutic vaccine. Nonetheless, given the inexistent prior knowledge on the mesothelioma immunopeptidome we considered that the exploration of the immunopeptidomics runs per se was an important matter. To address the reviewer's concerns, several figures have been merged, or part of the mentioned data has been completely removed or moved in the supplementary material in the updated version of the manuscript. Hopefully this updated version will be clearer to read and to follow.

1.6) Also, the figure 6 is confusing and unreadable, especially 6B. If the Gibbs cluster cannot generate the expected consensus binding motif, it might simply reflect that the quality of the immunopeptidome is insufficient to generate significant results.

We apologize for the lack of clarity in that particular section of the manuscript, and we appreciate the reviewer's feedback.

In the previously referred to as Figure 6 (which is now **updated as Figure 2**), our intention was to provide a direct comparison between two types of data visualizations for peptide-specificity deconvolution of the identified eluted peptides. Specifically, we aimed to compare the Gibbs cluster method (commonly used in immunopeptidomics analysis and quality control) with the stacked bar plot implemented in PyptidOmicsQC for MHC allele deconvolution.

Our observations revealed that when a low number of peptides were available for a given HLA or when different HLAs exhibited overlapping anchor positions, the Gibbs cluster allele deconvolution method proved to be more laborious and less immediate in demonstrating the HLA representation of the eluted peptides. In contrast, the stacked bar plot implemented in PyptidOmicsQC provided a more immediate and "quantitative" understanding of the HLA representation and coverage of the eluted peptides in each replicate. Furthermore, it illustrated that all biological replicates exhibited consistent patterns of peptide representation.

However, we have taken the reviewer's concern into account and have reworked the mentioned section of the manuscript to address this issue and improve the clarity of our explanation. Furthermore, we have made significant improvements to the figure based on the reviewer's instructions, resulting in a more readable version.

2. Fig 9A lacks many essential details to make any conclusion: What is the CTL only? What is the positive control (positive peptide control) in the experiment? In the method section, it's mentioned that the peptide concentration used was 10mM, which is way too high. Usually, for this type of assay, we tend to use a range of 10nM to 100 uM (Yang et al. Nature Medicine, 2019). This could explain the similar response induced by all the peptides tested. Also, a negative peptide control is mandatory to

demonstrate the specificity of the response. Flow cytometry results should also be shown with the minimum threshold (gating) established using a negative control as published elsewhere (Bear et al. Nature Communications 2021).

For figure 9B, the number of replicates is not reported and the SEM are so large that nothing can be concluded from these results. Also, do PMBCs from HD and MESO exhibit HLA subtypes matching the predicted binding of peptides tested? If so, it should be labeled in the figure with the flow cytometry gating results. Based on the results shown, there are no significant differences between HD and MESO. Therefore, the conclusion (lines 362-364) is over-stated.

We thank the reviewer for the comment as we had not realized the imperfections and typos present in this section.

In an attempt of strengthening our findings and with the aim to address other issues highlighted by the reviewers we completely remade this section from zero. Consequently, the addressed experiment shown in the Figure previously referred as **Figure 9A** has now been removed.

Also, the experiment shown in the Figure previously referred as **Figure 9B** has now been repeated with minor modifications to the protocol mentioned in the corresponding section of the material and methods. New data concerning the validation of the 10 selected human MPM-derived peptides is now presented in the **updated Figure 4** in the updated version of the manuscript.

3. The in vivo model used for the application of vaccine-based therapy seems to represent a good model since the immunopeptidome of AB12 cell line display a greater quality of MHC-I peptides. A table showing all the criteria used to filter the list of peptide candidates should be presented, like the log2FC of gene expression, predicted binding affinity by at least 2 algorithms, the peptide sequences and the HLA subtype predicted to bind the peptides. Additionally, a schematic of the bioinformatic pipeline used to end up with the 10 MPM mouse peptides selected from gene expression databases and cell lines with all the filtering criteria applied would add clarity for the reader.

We thank the reviewer for the insightful suggestion. In response to the reviewer's suggestion, we have generated a supplementary table that includes the requested information. This supplementary table is now referenced in the updated version of the manuscript, specifically in line 320.

Furthermore, we have incorporated schematics illustrating the peptide selection pipeline for subsequent immunological validation in both the murine and human aspects of our study. These schematics can be found in the updated Figures 5D and 4A, respectively.

We appreciate the reviewer's input, as it has contributed to enhancing the completeness and clarity of our manuscript

4. MHC class I expression at the cell surface of the MESO tumors from the patients should be quantified as performed with the other cell line. The results from this quantification could help to explain the poor quality of their corresponding immunopeptidomes.

This is indeed a very good advice, and we thank the reviewer for the comment. Unfortunately, there is no leftover material for us to run flowcytometry as all the material was directed to the immunopeptidomics. We will keep in mind for the next study.

5. It would be important to report mouse toxicity measurement (for instance weight measurement) for the in vivo model when monitoring the tumor growth under different experimental conditions.

We thank the reviewer for the comment; indeed, we explored the possible toxic effect of PeptiCRAd during a 5-weeks vaccination protocol. We did not observe any significant weight loss (toxicity) in mice vaccinated with PeptiCRAd for any of timepoints analyzed. See below.

6. A clearer hypothesis and rationale should be described and clarified in the introduction of the manuscript. In the abstract, the authors mentioned that MPM is responsive to immunotherapeutic cancers (line 44). However, in the introduction, the authors commented that ICI monotherapy has limited impact in MPM patients and ‘MPM are not ‘hot’ tumors, i.e. infiltrated with a of lot of T cells, but still infiltrated with T cells’. Moreover, the authors cited study reporting the low mutational burden of MPM. Therefore, the rationale of identifying new antigens and more importantly the origin of MHC-I peptides (self or non-self, mutated or over-expressed) should be mentioned in the beginning of the manuscript. The overview of the current knowledge of antigens associated to MPM should be described with the rationale. This type of cited studies would guide the reader to understand the selection of peptides based on transcriptomic and not exome sequencing data for instance.

We agree with the reviewer that both abstract and introduction needed additional work and clarification. We have extensively modified both sections in the updated version of the manuscript and, as suggested by the reviewer, we have in the revised introduction addressed the issue of the low mutational burden of mesothelioma and the need to focus on TAA as therapeutic targets rather than TSA. Additionally, we also sought to clarify that mesothelioma is generally infiltrated by T cells (although not ‘heavily’) and that immune checkpoint inhibitors have shown to be poorly effective as monotherapy, but they have shown some promise in improving patients’ responses when used as combination therapies.

References

Bauer, J., Köhler, N., Maringer, Y. et al. The oncogenic fusion protein DNAJB1-PRKACA can be specifically targeted by peptide-based immunotherapy in fibrolamellar hepatocellular carcinoma. *Nat Commun* 13, 6401 (2022). <https://doi.org/10.1038/s41467-022-33746-3>

Kevin A. Kovalchik et al. MhcVizPipe: A Quality Control Software for Rapid Assessment of Small- to Large-Scale Immunopeptidome Datasets, *Molecular & Cellular Proteomics*, Volume 21, Issue 1, 2022, 100178, ISSN 1535-9476, <https://doi.org/10.1016/j.mcpro.2021.100178>.

Yang W et al. Immunogenic neoantigens derived from gene fusions stimulate T cell responses. *Nat Med*. 2019 May;25(5):767-775. doi: 10.1038/s41591-019-0434-2. Epub 2019 Apr 22. PMID: 31011208; PMCID: PMC6558662.

Bear, A.S., Blanchard, T., Cesare, J. et al. Biochemical and functional characterization of mutant KRAS epitopes validates this oncoprotein for immunological targeting. *Nat Commun* 12, 4365

(2021). <https://doi.org/10.1038/s41467-021-24562-2>

Manuscript revised by:

Isabelle Sirois PhD

Senior scientist at CaronLab

1. J. Fritsche, D. J. Kowalewski, L. Backert, F. Gwinner, S. Dorner, M. Priemer, C.-C. Tsou, F. Hoffgaard, M. Römer, H. Schuster, O. Schoor, T. Weinschenk, Pitfalls in HLA Ligandomics—How to Catch a Li(e)gand. *Molecular & Cellular Proteomics* **20**, 100110 (2021).
2. T. J. O'Donnell, A. Rubinsteyn, U. Laserson, MHCflurry 2.0: Improved Pan-Allele Prediction of MHC Class I-Presented Peptides by Incorporating Antigen Processing. *Cell Systems* **11**, 42-48.e47 (2020).
3. A. Marcu, L. Bichmann, L. Kuchenbecker, D. J. Kowalewski, L. K. Freudenmann, L. Backert, L. Mühlenbruch, A. Szolek, M. Lübke, P. Wagner, T. Engler, S. Matovina, J. Wang, M. Hauri-Hohl, R. Martin, K. Kapolou, J. S. Walz, J. Velz, H. Moch, L. Regli, M. Silginer, M. Weller, M. W. Löffler, F. Erhard, A. Schlosser, O. Kohlbacher, S. Stevanović, H.-G. Rammensee, M. C. Neidert, HLA Ligand Atlas: a benign reference of HLA-presented peptides to improve T-cell-based cancer immunotherapy. *Journal for ImmunoTherapy of Cancer* **9**, e002071 (2021).
4. C. Capasso, M. Hirvonen, M. Garofalo, D. Romaniuk, L. Kuryk, T. Sarvela, A. Vitale, M. Antopolsky, A. Magarkar, T. Viitala, T. Suutari, A. Bunker, M. Yliperttula, A. Urtti, V. Cerullo, Oncolytic adenoviruses coated with MHC-I tumor epitopes increase the antitumor immunity and efficacy against melanoma. *Oncoimmunology* **5**, e1105429 (2016).
5. E. Ylösmäki, M. Fusciello, B. Martins, S. Feola, F. Hamdan, J. Chiaro, L. Ylösmäki, M. J. Vaughan, T. Viitala, P. S. Kulkarni, V. Cerullo, Novel personalized cancer vaccine platform based on Bacillus Calmette-Guèrin. *J Immunother Cancer* **9**, (2021).

REVIEWER COMMENTS

Reviewer #1 (Remarks to the Author):

The authors have addressed my concerns. However, there are still numerous minor changes to be made.

1. In the abstract, the authors wrote: "We characterized the immunogenicity profile of the eluted peptides in human healthy donors and cancer patients." It would be preferable to write: "We characterized in vitro the immunogenicity profile of the eluted peptides using T cells from human healthy donors and cancer patients."
2. in the results, the paragraph from line 157 to 163 referred to Figure 2A, but it appears to be supplementary fig1 B?
3. In the results, the end of the paragraph from line 168 to 173 should referred to Figure 1C.
4. Supplementary Table S1 is unreadable. The last letters of peptides sequence are cut. Furthermore, it would be interesting to add an additional column that give the putative HLA allele restriction.
5. Line 603, correct "peptidides"
6. Line 727, concentration of IL-2 and IL-15 should be added.
7. Line 283-285, killing by the MART1 control peptides stimulated T cell population is quite high on figure 4F for H28, perhaps due to allogenic response against H28 or MART-1 expression by H28. The authors should moderate their assertion for the killing of H28 on line 283-285.
8. line 831-840, where the virus alone or PC were injected to the mouse? subcutaneous injection?

Reviewer #2 (Remarks to the Author):

Overall, the authors thoughtfully incorporated the reviewer feedback, and I am supportive of manuscript publication.

I believe that the paragraph on line 591, section titled "MESO002 data cleanup" should be included. It may be helpful to replace "data" with immunopeptidome data" to be more descriptive if the character limit allows.

For Figure 1 legend- it may be helpful to include the name of HLA peptide prediction tools in the legend.

There is a typo in Figure 1 A. It should read "LC MS/MS" but it currently reads "LS MS/MS".

Reviewer #3 (Remarks to the Author):

The revised version submitted by Chiaro al. is significantly improved and most of the comments raised by the reviewers have been addressed. The addition of killing assays (Lactate and Luciferin assays) and the results showing a significant decrease of tumor growth in mice immunized with polyIC + peptides vs polyIC are convincing and strengthen the study. Some details in the new Figures 4 and 5 are lacking and importantly, positive controls are still required for the Elispot assays (see below).

General comments:

For Figure 4 panels B, C, D and E and Figure 5 panels G and H

♣ A positive control is required (such as Milteny Biotec's cytoStim) to show that PBMCs and CD8+ T cells from each healthy or patient sample have the ability to secrete IFN γ . Given that the results do not show a statistically significant difference, it is possible that some PBMC or CD8+ T cells do not respond. The use of positive controls would help to conclude on the qualitative effects observed for certain peptides. Similarly, some patients may respond non-specifically to any peptide and it is difficult to assess this with the data presented.

Furthermore, testing PBMCs from unmatched-HLA would provide a good negative control to

set up the minimal background of the assay. In general, up to 20 patients are needed to observe a statistical difference with a threshold of 20-25 spot-forming units (SPU). I

encourage the authors to read the article from Moodie, Z. et al.

(<https://link.springer.com/article/10.1007/s00262-010-0875-4>)

♣ Please provide the sequence of the peptides and their corresponding HLA subtype on the graphs (not only the number of the peptide 1, 2, 3, etc.).

♣ Please provide a representative Elispot image (unstimulated and with positive control) for each peptide tested for each of the panels Figures 4 and 5 discussed here.

Figure 4D and 4F

♣ Figure 4D doesn't show any significant results or difference between MIX A and MIX B with CD8+T cells from healthy donors. However, differences are observed in mesothelioma cells in figure F between Mix A and MIX B and the term killing capacity is used. The test used in panel F is not clear as compared to the lactate or luciferin assays used to define killing capacity in other figure's panels. Please clarify the use of this test and how the panels D and F are related.

Specific comments:

Figure 4A

♣ The schematic generated to illustrate the selection of peptides is very useful. It would be more informative to add the following details inside the rectangles:

o Absolute number of peptides following application of the criteria

o The name of the database used for GE analysis as well as the threshold of FC and p-value applied to select peptides

o Please specify the word 'Found'. What is the minimum number of cell lines or patients needed to select peptides?

Figure 4E

♣ It's not clear the difference between the 2 graphs in this panel. Please specify or re-label the graph.

Figure 4F

- ♣ For visual clarity, label the Y axis with the same scale values.
- ♣ It would be important to mention the number of biological replicates and provide some statistical testing in the caption of the figure.
- ♣ It would be important to define in the caption what means 'Target Alone'.

Figure 5D

- ♣ Same as requested for Figure 4A.

Isabelle Sirois PhD

REVIEWER COMMENTS

See below the document containing our answers to the reviewers' comments. The reviewers' concerns are presented in bold, while our answers are shown below each one of them, in regular font.

Reviewer #1 (Remarks to the Author):

The authors have addressed my concerns. However, there are still numerous minor changes to be made.

A: We thank the reviewer for the constructive criticism, and we are happy to realize that we have been able to address all the reviewer's concerns about our work.

1. In the abstract, the authors wrote: "We characterized the immunogenicity profile of the eluted peptides in human healthy donors and cancer patients." It would be preferable to write: "We characterized in vitro the immunogenicity profile of the eluted peptides using T cells from human healthy donors and cancer patients."

A: We thank the reviewer for the suggestion which clarify our message. We have modified the text as indicated in lines 56-57

2. in the results, the paragraph from line 157 to 163 referred to Figure 2A, but it appears to be supplementary fig1 B?

A: We thank the reviewer for pointing this out. The mistake has been corrected in the updated version of the manuscript in line 167

3. In the results, the end of the paragraph from line 168 to 173 should referred to Figure 1C.

A: We thank again the reviewer for pointing out this mistake. The text has been modified in the updated version of the manuscript in line 177.

4. Supplementary Table S1 is unreadable. The last letters of peptides sequence are cut. Furthermore, it would be interesting to add an additional column that give the putative HLA allele restriction.

A: We apologize for the inconvenience. A new version of the table S1 table has been produced following the reviewers' comments.

5. Line 603, correct "peptidides"

A: the spelling mistake has been corrected in line 608.

6. Line 727, concentration of IL-2 and IL-15 should be added.

A: We thank the reviewer for pointing out something we have overlooked. The concentrations of IL2 and IL15 have been added to the method section of the updated version of the manuscript in line 732-733. Specifically, the concentration of IL-2 used was 50U/ml, while for IL-15 we used a concentration of 10ng/ml.

7. Line 283-285, killing by the MART1 control peptides stimulated T cell population is quite high on figure 4F for H28, perhaps due to allogenic response against H28 or MART-1 expression by H28. The authors should moderate their assertion for the killing of H28 on line 283-285.

A: We agree with the reviewer that allogenic response represented an issue in our experimental assay. We believe that this effect looks far more evident in the experiment involving H28 compared to the H2452 as these latter were proliferating more rapidly.

However, despite the effect of allogenic killing, significant higher cell killing was observed for T cells expanded using either MIX A or MIX B compared to T cells expanded using MART1. We also observed that T cells expanded using MIX A showed a better killing capacity compared to cells expanded using MIX B. We hope that the new visualization we propose in the updated figure 4F could better convey these insights. Additionally, the text regarding this figure was moderated according to the reviewer's suggestion in the updated version of the manuscript in lines 287-291.

8. line 831-840, where the virus alone or PC were injected to the mouse? subcutaneous injection?

A: We thank the reviewer for pointing this out. All the priming and boosting injections for the immunization of mice using PeptiCRAAd were performed subcutaneously. We have updated the text in the new version of the manuscript in line 842- 843.

Reviewer #2 (Remarks to the Author):

Overall, the authors thoughtfully incorporated the reviewer feedback, and I am supportive of manuscript publication.

A: We thank the reviewer for the comments, and we are happy to have addressed all the reviewer's concerns and we greatly appreciate the acknowledgment of our hard work.

I believe that the paragraph on line 591, section titled "MESO002 data cleanup" should

be included. It may be helpful to replace "data" with immunopeptidome data" to be more descriptive if the character limit allows.

A: We thank the reviewer for providing valuable feedback. As per the reviewer's suggestion, we have made adjustments to the title of the method section in line 596 of the revised manuscript. Furthermore, within the corresponding "Results" section (lines 154 to 161), we have incorporated additional text to elucidate the utilization of an in silico clean-up approach for the immunopeptidomic dataset whose methodology is further elaborated upon later in the methods section.

For Figure 1 legend- it may be helpful to include the name of HLA peptide prediction tools in the legend.

A: We thank the reviewer for bringing this to our attention. We acknowledge that this information was missing and apologize for the oversight. In the updated version of Figure 1 legend, we have now included the name of the MHC-binding affinity tool used in our study.

There is a typo in Figure 1 A. It should read "LC MS/MS" but it currently reads "LS MS/MS".-

A: We thank the reviewer for pointing this typo out and we apologize for the oversight. Figure 1A infographic has now been corrected and it now reads "LC MS/MS".

Reviewer #3 (Remarks to the Author):

The revised version submitted by Chiaro al. is significantly improved and most of the comments raised by the reviewers have been addressed. The addition of killing assays (Lactate and Luciferin assays) and the results showing a significant decrease of tumor growth in mice immunized with polyIC + peptides vs polyIC are convincing and strengthen the study. Some details in the new Figures 4 and 5 are lacking and importantly, positive controls are still required for the Elispot assays (see below).

A: We appreciate the reviewer for recognizing the effort we put into addressing the previously underlined shortcomings of our study.

General comments:

For Figure 4 panels B, C, D and E and Figure 5 panels G and H

- **A positive control is required (such as Milteny Biotec's cytoestim) to show that PBMCs and CD8+ T cells from each healthy or patient sample have the ability to secrete IFN γ . Given that the results do not show a statistically significant difference,**

it is possible that some PBMC or CD8+ T cells do not respond. The use of positive controls would help to conclude on the qualitative effects observed for certain peptides. Similarly, some patients may respond non-specifically to any peptide and it is difficult to assess this with the data presented. Furthermore, testing PBMCs from unmatched-HLA would provide a good negative control to set up the minimal background of the assay. In general, up to 20 patients are needed to observe a statistical difference with a threshold of 20-25 spot-forming units (SPU). I encourage the authors to read the article from Moodie, Z. et al.

<https://link.springer.com/article/10.1007/s00262-010-0875-4>

A: We appreciate the reviewer's suggestion. In our ELISpot assay, we included positive controls to detect both biological issues, such as interferon-gamma secretion in PBMC batches, and technical issues related to interferon-gamma detection. For this purpose, we utilized eBioscience™ Cell Stimulation Cocktail (500X) (Cat # 00-4970-93), a mix of PMA and ionomycin. However, when used at the recommended concentration by the provider, we observed a complete blue well, making spot counting impossible. As a result, we can only detect severe assay issues (on/off) and unfortunately, we cannot use the spot counts to fine-tune or normalize our data. Instead, for data normalization we can solely rely on the number of PBMCs seeded at day 0 of the assay. For the future assays, we will surely try the suggested approach and reagent.

Regarding the use of HLA-unmatched peptides as negative controls for the ELISpot assay, we agree that it could be beneficial. However, we chose not to use them due to two main reasons: Firstly, guaranteeing no possible cross-presentation across all subjects involved in the test would be challenging, and these peptides might trigger unspecific T cell activation. Secondly, given our focus on unmutated tumor-associated antigens, we anticipated a low response magnitude, hence, we preferred to maximize the number of seeded PBMCs per well for each of our test peptides.

Despite not utilizing an HLA-mismatched peptide as a negative control in this study, we still observed significant variation in the T cell activation among the set of different peptides we tested. We hope that this is sufficient to support our claim that some of the peptides indeed are immunogenic.

We appreciate the reviewer's keen eye for detail. Upon careful consideration, we realized that we may have used too few replicates (two or three wells) or too few subjects to draw irrefutable conclusions. Our study's main challenge lies in the low response expected from the selected peptides in both healthy donors, who are immunologically naïve, and patients, particularly elderly subjects with expected lower T cell activity. To address this, we adopted the threshold identified by Rojas et al. (1), where they established a threshold of 7 spots per 300,000 PBMCs in pre-vaccinated healthy donors. We normalized this value on 1'000'000 PBMCs and draw the threshold in the figures which required it.

Furthermore, in the revised figure, we have included additional statistical information to better convey our intended message. We hope this clarifies our approach and findings.

- Please provide the sequence of the peptides and their corresponding HLA subtype on the graphs (not only the number of the peptide 1, 2, 3, etc.).

A: We agree with the reviewer that this modification will improve the understanding and readability of Figure 4 and 5. Following the reviewer’s suggestion, we included a table containing peptide sequence and the corresponding HLA specificity in the new version of the Figure 4 and 5.

- Please provide a representative ELISpot image (unstimulated and with positive control) for each peptide tested for each of the panels Figures 4 and 5 discussed here.

Please find below collections of representative ELISpot images of the data showed in both figures 4 and 5.

*Due to the insufficient number of available cells, cells were not to seed for the positive control

Collection of representative ELISpot images of the data showed in Figure 4. As for the ELISpot referring to the data shown in Figure 4E, due to insufficient number of cells after T cell expansion with tumor lysate for the subject BC-RC2, we decided not to seed cells for the positive control.

Collection of representative ELISpot images of the data showed in Figure 5. As for the ELISpot images referring to Figure 5G mice were divided in two immunization groups for practical reasons. The two immunization groups were immunized at different times, but the data are presented altogether in Figure 5. Each immunization group contains its own Mock and Poly IC (Adjuvant alone) groups.

Figure 4D and 4F

- Figure 4D doesn't show any significant results or difference between MIX A and MIX B with CD8+T cells from healthy donors. However, differences are observed in mesothelioma cells in figure F between Mix A and MIX B and the term killing capacity is used. The test used in panel F is not clear as compared to the lactate or luciferin assays used to define killing capacity in other figure's panels. Please clarify the use of this test and how the panels D and F are related.**

A: In Figure 4D we wanted to show that we managed to obtain a T cell reactivity (stronger or weaker) towards most of the peptides employed for the stimulation. However, we still observed heterogeneous responses between subjects.

Ultimately, after T cell expansion we observed that T cells expanded with the mix of peptides A or B (MIX A, MIX B) were overall better in killing mesothelioma cell lines H28 or H2452 compared to T cells expanded using the control peptide MART1. We agree with the reviewer regarding the clarity of Figure 4F, and we have modified it to match the killing assays shown in Figure 5 (LDH) and Figure 6 (Luciferin). We hope that this way the data are conveyed in a clearer fashion. Additional information on the statistical test used has been added to the figure legend.

Specific comments:

Figure 4A

- **The schematic generated to illustrate the selection of peptides is very useful. It would be more informative to add the following details inside the rectangles:**
 - **Absolute number of peptides following application of the criteria**
 - **The name of the database used for GE analysis as well as the threshold of FC and p-value applied to select peptides**
 - **Please specify the word 'Found'. What is the minimum number of cell lines or patients needed to select peptides?**

A: We agree with the reviewer that adding this information would clarify the figure. Hence, we present below the revised infographic concerning peptide selection, which integrates the reviewer's suggestions.

To provide a comprehensive understanding, we have introduced a clear elucidation of "manual curation" within the figure legend. In essence, this process involved taking into account additional criteria, such as peptide identification across one or more cell lines or the presence of the source gene in the list of genes upregulated in mesothelioma, as outlined in the work by Barone et al. This particular study is referenced in the main text of our manuscript.

Regrettably, the peptides that met all of the established criteria were fewer than our intended goal of 10. Consequently, we adopted a more flexible approach with the last selection criteria, which fall under the umbrella of the "manual curation".

Figure 4E

- **It's not clear the difference between the 2 graphs in this panel. Please specify or re-label the graph.**

A: Figure 4E represents 2 independent experiments conducted using different healthy donors' CD8+ T cells. Specifically, the donors showed are BC-RC2 and BC-RC3. We observed that tumor lysates were able to expand peptide specific T cells but in different amount depending on the donor. We speculate this phenomenon to be due to the initial number of peptide specific T cells in each sample. To clarify the figure, the graphs have been re-labelled.

Figure 4F

- **For visual clarity, label the Y axis with the same scale values.**

A: Unfortunately, we are afraid that with the addition of new data this modification would result in a lower the clarity for the reader.

- **It would be important to mention the number of biological replicates and provide some statistical testing in the caption of the figure.**

A: We performed two independent experiments with T cells derived from 2 donors. The Statistical test used was an ordinary one-way ANOVA with Fisher LSD test. The additional information required has been added to the figure legend.

- **It would be important to define in the caption what means 'Target Alone'.**

A: We agree with the reviewer that this terminology was rather confusing. Upon further consideration we decided to remove the column corresponding to the "Target alone". These latter corresponded to the scenario where either of the mesothelioma cell lines H28 or H2052 were seeded without the addition of any effector cell (T cells). We believe that the focus on the comparison with the "negative control", represented in the use of T cells expanded in presence of the MART-1 peptide, better conveys the results.

Figure 5D

- **Same as requested for Figure 4A.**

A: We, once again, agree with the reviewer that adding this information would clarify the infographic shown in Figure 5. Therefore, we present below the new infographic regarding the murine peptides' selection, which has been updated following the reviewer's suggestions.

A clear explanation of the “manual curation” has been added to the corresponding figure legend. Briefly, the 15 candidates which seemed to best fulfill the selected criteria were chosen for the following immunological validation.

REVIEWERS' COMMENTS

Reviewer #3 (Remarks to the Author):

Chiaro, J. et al' s answered all the points raised by the reviewer. I would like to insist on adding all the ELISA spot images provided in the rebuttal letter for figures 4 and 5 as supplementary data in the manuscript. These results are very useful for the reader. With these additions, I agree that the manuscript should be considered for publication.

Isabelle Sirois PhD

Response to the reviewer's comment

The reviewer comment is presented in **bold** while the answer is presented in normal lettering.

REVIEWERS' COMMENTS

Reviewer #3 (Remarks to the Author):

Chiaro, J. et al' s answered all the points raised by the reviewer. I would like to insist on adding all the ELISA spot images provided in the rebuttal letter for figures 4 and 5 as supplementary data in the manuscript. These results are very useful for the reader. With these additions, I agree that the manuscript should be considered for publication.

Isabelle Sirois PhD

A: We would like to thank the reviewer for the rounds of revisions as the comments raised helped improving the quality of our work. As for reviewer's request, the images of the ELISpot concerning the figures 4 and 5 are provided as supplementary data in the updated version of the manuscript.